# STRUCTURAL INFERENCE WITH DYNAMICS ENCODING AND PARTIAL CORRELATION COEFFICIENTS

**Aoran Wang** [1] & **Jun Pang** [1,2]
[1] Faculty of Science, Technology and Medicine, University of Luxembourg
[2] Institute for Advanced Studies, University of Luxembourg
`{aoran.wang, jun.pang}@uni.lu`

## ABSTRACT

This paper introduces a novel approach to structural inference, combining a variational dynamics encoder with partial correlation coefficients. In contrast to prior methods, our approach leverages variational inference to encode node dynamics within latent variables, and structural reconstruction relies on the calculation of partial correlation coefficients derived from these latent variables. This unique design endows our method with scalability and extends its applicability to both one-dimensional and multi-dimensional feature spaces. Furthermore, by reorganizing latent variables according to temporal steps, our approach can effectively reconstruct directed graph structures. We validate our method through extensive experimentation on twenty datasets from a benchmark dataset and biological networks. Our results showcase the superior scalability, accuracy, and versatility of our proposed approach compared to existing methods. Moreover, experiments conducted on noisy data affirm the robustness of our method.

## 1 INTRODUCTION

Complex dynamical systems often find their mathematical representations as networks of interacting agents. In this conceptualization, the system can be visualized as an interaction graph, where individual agents are nodes, their interactions are edges, and the underlying system's structure is encapsulated within an adjacency matrix. This modeling paradigm transcends diverse domains, finding applications in understanding physical systems (Kwapień & Drożdż, 2012; Ha & Jeong, 2021), unraveling multi-agent dynamics (Brasó & Leal-Taixé, 2020; Li et al., 2022), and deciphering the intricate interplay within biological systems (Tsubaki et al., 2019; Pratapa et al., 2020). As the pursuit of understanding dynamical systems gains momentum, there arises an imperative—unveiling the structure concealed within the intricate web of interactions. The recognition of this hidden framework not only advances our comprehension of the system's intrinsic mechanisms but also empowers us to predict and control its behavior, a goal of paramount significance. However, a common challenge prevails across these scenarios: the availability of only a fraction of nodes' observable features within a limited temporal window. This limitation leaves the underlying structural skeleton partially or completely veiled by the complex dynamical processes at play. To surmount this challenge, a need arises for an approach capable of reconstructing the concealed structure of dynamical systems solely from the observable features.

We introduce the notion of *a trajectory*, which amalgamates observed features from all nodes over a defined time interval. These trajectories encapsulate a rich tapestry of information, encoding the evolution of node features—a result of their historical states and interactions with fellow agents. Several methodologies grounded in variational autoencoders (VAEs) have emerged with the aim of unearthing the underlying structure based on these trajectories (Kipf et al., 2018; Alet et al., 2019; Webb et al., 2019; Chen et al., 2021; Löwe et al., 2022; Wang & Pang, 2022; Wang et al., 2023a). Guided by the Information Bottleneck (IB) principle, they navigate the reconstruction of dynamical systems' adjacency matrices within their latent spaces. Yet, a substantial limitation looms large: these approaches mandate a global perspective of the entire system, rendering structural inference on large-scale graphs computationally infeasible. In contrast, correlations and partial correlations offer an alternative route to structure recovery, demonstrating scalability irrespective of graph size (Whit-

taker, 1990; Pratapa et al., 2020). However, they require the node features to be one-dimensional at every time step and lack the versatility to decipher directed graph structures.

In response to these challenges, we introduce a novel structural inference framework that harnesses the strengths of VAEs and partial correlations, **S**tructural **I**nference with **D**ynamics **E**ncoding and Partial **C**orrelation Coefficients (SIDEC). Leveraging a VAE variant known as the variational dynamics encoder (VDE) (Hernández et al., 2018), we encode node feature dynamics over a defined temporal window into latent variables. Concurrently, we reduce feature dimensionality by extracting essential information—creating a versatile foundation for structural inference. These latent variables form the basis for calculating partial correlation coefficients (PCOR), enabling the reconstruction of underlying interaction graph structures. SIDEC transcends the limitations of its predecessors, enabling structural inference on extensive graphs without an exponential surge in computational demands. It extends its applicability to trajectories featuring both one-dimensional and multi-dimensional node features. Remarkably, through the reorganization of latent variables, temporal order information is extracted, enabling precise inference of directed graph structures. Experimental results validate the prowess of SIDEC, surpassing previous methods in both accuracy and scalability, while demonstrating resilience to diverse noise conditions.

Our contributions encompass the following aspects:

- We propose a novel structural inference approach, which is rooted in variational dynamics encoder and partial correlation coefficients.
- This method effectively addresses scalability concerns, does not need a prior for structure, and provides a versatile solution for structural inference across various feature dimensions.
- Through innovative latent variable manipulation, we expand its capabilities to accurately infer directed graph structures.
- Empirical validation underscores its superiority over existing benchmarks and its robustness in the presence of additive Gaussian noise.

## 2 RELATED WORK

**Structural inference with VAEs.** Structural inference seeks to faithfully reconstruct the interaction graph underlying a dynamical system, relying solely on observational data representing the state of agents over time. Neural Relational Inference (NRI) (Kipf et al., 2018) is a seminal work that ventured into structural inference, employing VAEs within a fixed fully connected graph framework. Building on the foundation laid by NRI, subsequent research has expanded the horizons of this field. These extensions encompass the adaptation to multi-interaction systems (Webb et al., 2019), the incorporation of efficient message-passing mechanisms (Chen et al., 2021), the exploration of modular meta-learning (Alet et al., 2019), the iterative elimination of indirect connections (Wang & Pang, 2022), and the development of efficient structural inference techniques integrated with reservoir computing networks (Wang et al., 2023a). Despite these advancements, a common limitation among these methods is their reliance on latent spaces to infer entire graph structures, which constrains their scalability, particularly when confronted with larger graphs. Moreover, their loss functions naturally introduce a uniform prior on the structure, which is not versatile for all graphs.

**Structural inference with correlations and partial correlations.** In addition to VAE-based structural inference approaches, researchers have harnessed correlation and partial correlation methodologies to unravel the structures of interaction graphs. These techniques encompass diverse strategies, including Pearson correlation (Maucher et al., 2011; Specht & Li, 2017), distance correlation (Liu et al., 2021), partial correlations (Papili Gao et al., 2018; Millington & Niranjan, 2019), low-order partial correlations (Zuo et al., 2014), and semi-partial correlation coefficients (Kim, 2015). While these methods excel within specific domains, such as gene regulatory networks and financial networks, they grapple with limitations. Notably, they struggle to accommodate multi-dimensional node features and lack the capability to discern the directionality of edges. To address these challenges, this work introduces a fresh perspective by repurposing VAEs to shift their focus from structural inference to encoding node dynamics with the help of VDE. By integrating partial correlation calculations into the reconstruction process, this approach overcomes scalability challenges and ensures robustness across various node feature dimensionalities.

## 3   PRELIMINARIES

In this section, we provide preliminaries and background knowledge.

**Notations and problem formulation.** We conceptualize a dynamical system as a directed underlying interaction graph, where the system's agents correspond to nodes, and the directed interactions among these agents manifest as edges in the graph. Denoted as $\mathcal{G} = (\mathcal{V}, \mathcal{E})$, this directed graph comprises $\mathcal{V}$, the feature set of $n$ nodes, and and $\mathcal{E}$, the set of edges. The temporal evolution of nodes' features is encapsulated in trajectories: $\mathcal{V} = \{V\} = \{V^0, V^1, \ldots, V^{T-1}\}$, spanning $T$ time steps, with $V^t$ signifying the feature set of all $n$ nodes at time step $t$: $V^t = \{v_0^t, v_1^t, \ldots, v_{n-1}^t\}$. The feature vector at time $t$ for node $i$, denoted as $v_i^t \in \mathbb{R}^d, 0 \le t \le T-1$, is $d$-dimensional. Usually, a set of $M$ trajectories is observed: $\{V_{[1]}, V_{[2]}, \ldots, V_{[m]}, \ldots, V_{[M]}\}$. We assume that the nodes are observed in their entirety, and $\mathcal{E}$ remains immutable during the observation. From $\mathcal{E}$, we derive an asymmetric adjacency matrix denoted as $\mathbf{A} \in \mathbb{R}^{n \times n}$. Within $\mathbf{A}$, each element $\mathbf{a}_{ij} \in \{0, 1\}$ indicates the presence ($\mathbf{a}_{ij} = 1$) or absence ($\mathbf{a}_{ij} = 0$) of an directed edge from node $i$ to node $j$. The dynamics of the system are significantly influenced by $\mathbf{A}$, with node $i$'s features at time $t+1$ as:

$$v_i^{t+1} = v_i^t + \Delta \cdot \sum_{j \in \mathcal{U}i} f(||v_i, v_j||_\alpha), \tag{1}$$

where $\Delta$ denotes a time interval, $\mathcal{U}\_i$ is the set of nodes connected with node $i$, and $f(\cdot)$ is the state-transition function deriving to dynamics caused by the edge from node $j$ to $i$, and $||\cdot, \cdot||\_\alpha$ denotes the $\alpha$-distance. Importantly, $\mathcal{U}\_i$ is derived from the adjacency matrix $\mathbf{A}$, highlighting the role of $\mathbf{A}$ in determining the interactions between nodes and the system's dynamics. This paper primarily focuses on the challenge of *structural inference*, which involves the unsupervised reconstruction of the structure encapsulating the underlying interaction graph based on $M$ observable sets of trajectories.

**Information bottleneck.** The *Information Bottleneck* (IB) theory has emerged as a valuable framework for understanding the inner workings of deep neural networks (Tishby et al., 1999; Tishby & Zaslavsky, 2015; Shwartz-Ziv & Tishby, 2017). At its core, IB aims to discover a minimal sufficient representation, denoted as $Z$, for a given input data $X$ and its label $Y$. This discovery is achieved by minimizing the expression $I(Z; X) - \mathfrak{u} \cdot I(Z; Y)$, where $\mathfrak{u}$ serves as a Lagrangian multiplier that balances the trade-off between sufficiency and minimality. Furthermore, the Variational Information Bottleneck (VIB) (Alemi et al., 2017), introduces a variational approximation to the IB theory. It unveils that the objective function of VAEs constitutes a specific instance of this approximation. In the realm of unsupervised learning, VAEs excel at extracting the minimal sufficient statistics from input features, facilitating the derivation of output features while retaining compressed and abstract representations within their latent spaces.

Previously, VAE-based structural inference techniques (Kipf et al., 2018; Webb et al., 2019; Chen et al., 2021; Wang & Pang, 2022; Wang et al., 2023a) have employed the VIB framework to reconstruct the adjacency matrix within the VAE's latent space, expressed as:

$$\mathbf{Z} = \arg \min_{\mathbf{Z}} I(\mathbf{Z}; V^t, \mathbf{A}) - \mathfrak{u} \cdot I(\mathbf{Z}; V^{t+1}). \tag{2}$$

Initially, these methods assume a fully connected graph $\mathbf{A}$ and subsequently sample $\hat{\mathbf{A}}$ from the latent space $\mathbf{Z}$. However, scalability issues arise when confronted with graphs exceeding 100 nodes, as they require learning representations for each node pair, necessitating processing of the entire graph. Additionally, the assumption of a uniform distribution in the Kullback-Leibler (KL) term of the loss function implies equal edge existence probabilities, an assumption unsuitable for all graphs. In contrast, our approach, SIDEC, harnesses the power of a Variational Dynamics Encoder (VDE) (Hernández et al., 2018). It conceptualizes the minimal sufficient statistics within the latent space as dynamics within the node features, offering both scalability and the elimination of assumptions regarding edge existence probabilities.

**Partial correlations.** Partial correlation is a statistical concept that quantifies the association between two random variables after removing the influence of all other random variables (Kim, 2015). In the context of this research, it serves to estimate the direct relationship or association between two nodes, essentially indicating the existence of interaction between them. Let's consider a random vector $X = (x_1, x_2, \ldots, x_i, \ldots, x_n)'$ where $|X| = n$. We denote the variance of a variable random $x_i$ as $Var_i (= var(x_i))$ and the covariance between two random variables $x_i$ and $x_j$ as $Cov_{ij} (= cov(x_i, x_j))$. The variance-covariance matrices of random vectors

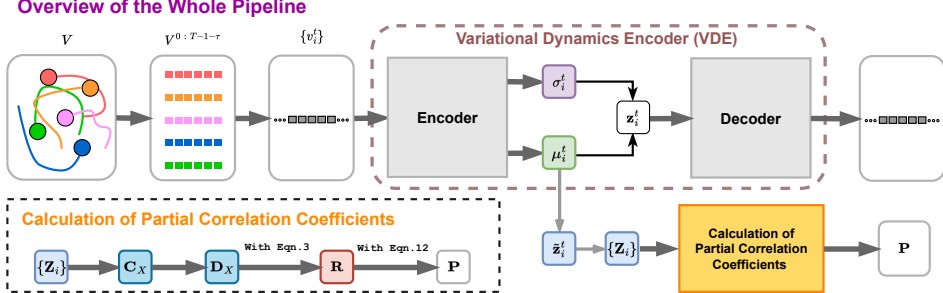

Figure 1: The pipeline of SIDEC. The wrapped box on the bottom left shows the pipeline of the calculation of partial correlation coefficients (PCOR).

$X$ and $X_S$ ($S \subset \{1, 2, ..., n\}$ and $|S| < n$) are denoted as $C_X$ and $C_S$, respectively, where $X_S$ is a random sub-vector of $X$. The correlation between two random variables $x_i$ and $x_j$ is $r_{ij} = \frac{Cov_{ij}}{\sqrt{Var_i}\sqrt{Var_j}} (= corr(x_i, x_j))$. The partial correlation $r_{ij|k}$ of $x_i$ with $x_j$ given $x_k$ is:

$$r_{ij|k} = \frac{r_{ij} - r_{ik}r_{jk}}{\sqrt{1 - r_{ik}^2}\sqrt{1 - r_{jk}^2}}, \tag{3}$$

For cases involving more than three variables, such as more than three nodes in a graph, we extend the formulation to higher-order partial correlations. This involves considering the inverse variance-covariance matrix of $X$, denoted as $D_X = C_X^{-1}$. Here, $d_{ij}$ and $c_{ij}$ are the $(i, j)$-th elements of the matrices $D_X$ and $C_X$, respectively. Then following (Whittaker, 1990), the partial correlation of $x_i$ with $x_j$ given a random vector $X_S$ is:

$$r_{ij|S} = -\frac{d_{ij}}{\sqrt{d_{ii}}\sqrt{d_{jj}}}. \tag{4}$$

$X_S$ is the random sub-vector of $X$ after removing the random variables $x_i$ and $x_j$. Its size is $|S| = |X| - 2$. For more details about the derivation, please refer to Appendix A.1. While Eqn. 4 provides insights into the graph's structure, it should be noted that partial correlation alone is insufficient. The methodology for obtaining the graph's structure is discussed in Section 4.3.

## 4 METHOD

### 4.1 ENCODING DYNAMICS WITH VARIATIONAL DYNAMICS ENCODER

Unlike previous VAE-based structural inference methods (Kipf et al., 2018; Webb et al., 2019; Chen et al., 2021; Wang & Pang, 2022; Wang et al., 2023a), which employ VAEs to simultaneously learn edge representations and reconstruct the adjacency matrix of the graph in their latent spaces, SIDEC takes a different approach. It leverages a Variational Dynamics Encoder (VDE) (Hernández et al., 2018) designed to capture minimal information about the dynamics of node features within a defined time window. While the architecture of VDE bears some resemblance to a VAE, it is tailored for time-series data. The VDE is presented in the top row in Fig. 1. Specifically, VDE predicts the state of a node ($v_i^{t+\tau}$) at a future time step ($t + \tau$) based on its current state ($v_i^t$), where $\tau$ represents the length of the time window required for the system's dynamics to be considered Markovian.

To clarify, prior VAE-based methods aim to preserve minimal sufficient statistics in the latent space and formulate it as the adjacency matrix $\hat{\mathbf{A}}$ of the graph, as depicted in Eqn. 2. In contrast, SIDEC employs VDE to learn minimal sufficient statistics essential for encoding node feature dynamics within a specific time window, which can be expressed through the VIB principle:

$$\mathbf{z}_i^t = \underset{\mathbf{z}_i^t}{\arg\min} \, I(\mathbf{z}_i^t; v_i^t) - \mathfrak{u} \cdot I(\mathbf{z}_i^t; v_i^{t+\tau}), \tag{5}$$

where $v_i^t$ is the feature of node $i$ at time $t$, and $v_i^{t+\tau}$ is its feature after a time window $\tau$. The information about the dynamics is encoded in the latent space of VDE.

**Implementation.** Suppose we have a set of $M$ observable trajectories of a dynamical system with $n$ nodes, with each trajectory covering the evolution of all node features within a time period $T$. At each time step, VDE processes the features of a single node. VDE comprises three operations: an

encoder parameterized by $\phi$ that computes $q_\phi(\mathbf{z}_i^t|v_i^t)$, a decoder parameterized by $\theta$ that computes $p_\theta(v_i^{t+\tau}|\mathbf{z}_i^t)$, and a reparametrization step. The implementation of VDE can be formulated as:

$$\textbf{Encoder:} \qquad \mu_i^t, \sigma_i^t = \text{ENC}(v_i^t), \qquad (6)$$

$$\textbf{Reparametrization:} \qquad \mathbf{z}_i^t = \mu_i^t + \sigma_i^t \odot \epsilon, \text{ where } \epsilon \sim \mathcal{N}(0, \mathbf{I}), \qquad (7)$$

$$\textbf{Decoder:} \qquad \hat{v}_i^{t+\tau} = \text{DEC}(\mathbf{z}_i^t), \qquad (8)$$

where $v_i^t$ is the input node feature at time $t$, $\hat{v}_i^{t+\tau}$ is the prediction of node feature after a time window $\tau$. ENC and DEC are encoder and decoder operations, respectively, and are implemented as multi-layer perceptrons (MLPs). For the exact implementation of the encoder and decoder please refer to Appendix A.7. $\mu_i^t$ and $\sigma_i^t$ are the outputs of the encoder, with $\odot$ as the element-wise production. To reduce the dimensionality for PCOR calculations, the scale of $\mathbf{z}_i^t$ to be one. Importantly, $\tilde{\mathbf{z}}_i^t = \mu_i^t$, ensuring that the learned variable $\tilde{\mathbf{z}}_i^t$ encodes the dynamics for node features from time $t$ to $t + \tau$.

**Advantages.** Leveraging VDE to encode node dynamics offers several advantages:

1. **Focus on dynamics:** By adhering to the VIB principle, VDE concentrates on capturing minimal sufficient statistics that encode the dynamics of node features, making it particularly well-suited for modeling dynamical systems.
2. **Scalability:** Unlike previous VAE-based methods, VDE operates solely on node features and doesn't necessitate a complete view of the entire $n$-node system, enhancing scalability.
3. **No uniform prior:** VDE doesn't impose a uniform prior on the edges, unlike previous methods, making it more adaptable to various graph types.
4. **Dimensionality reduction:** VDE can effectively reduce the dimensions of node features, accommodating dynamical systems with varying feature counts and enabling the use of correlation calculations, which inherently require one-dimensional features.

## 4.2 TRAIN WITH A HYBRID LOSS

In this study, the training of the VDE hinges on a composite loss function comprising two key components: the reconstruction loss, denoted as $\mathcal{L}_{\text{recon}}$, and the autocorrelation loss, represented as $\mathcal{L}_{\text{autoc}}$. The reconstruction loss, $\mathcal{L}_{\text{recon}}$, derives inspiration from the widely employed Evidence Lower Bound loss for VAEs (Kingma & Welling, 2013). It comprises two integral segments: one that evaluates the prediction accuracy of node features after a time window $\tau$, and another that assesses the KL divergence between latent space priors. Formally expressed as:

$$\mathcal{L}_{\text{recon}} = \mathbb{E}_{q_\phi(\mathbf{z}|v^t)}[\log\ p_\theta(v^{t+\tau}|\mathbf{z})] - D_{KL}(q_\phi(\mathbf{z}|v^t) \parallel p_\theta(\mathbf{z})). \qquad (9)$$

This loss function is implemented as:

$$\mathcal{L}_{\text{recon}} = \frac{1}{L}\sum \|\hat{v}^{t+\tau} - v^{t+\tau}\|^2 - \frac{1}{2L}\sum(1 + \log\sigma^2 - \sigma^2 - \mu^2), \qquad (10)$$

where $L$ signifies the batch size. The autocorrelation loss, $\mathcal{L}_{\text{autoc}}$, serves the purpose of optimizing VDE to capture comprehensive representations of long-time kinetics within time-series data (Hernández et al., 2018). This loss seeks to ensure that no process in the data evolves more slowly than the true underlying process, as proposed by Noé & Nuske (2013). A favorable outcome of this optimization process results in phase-space decomposition that yields a linear model with larger leading dynamical eigenvalues, enhancing the overall representation quality. For a single-layer decomposition of phase-space, equivalent to the autocorrelation of $\mu$ (Noé & Nuske, 2013), the autocorrelation loss is defined as:

$$\mathcal{L}_{\text{autoc}} = -\frac{\mathbb{E}[(\mu^t - \bar{\mu}^t)(\mu^{t+\tau} - \bar{\mu}^{t+\tau})]}{\mathbf{s}_{\mu^t}\mathbf{s}_{\mu^{t+\tau}}}, \qquad (11)$$

where $\bar{\mu}$ and $\mathbf{s}_\mu$ are the sample mean and population mean of the latent space vector for a specific data batch. Consequently, the combined loss for training VDE is formulated as:

$$\mathcal{L}_{\text{VDE}} = \mathcal{L}_{\text{recon}} + \mathcal{L}_{\text{autoc}}. \qquad (12)$$

## 4.3 STRUCTURAL INFERENCE WITH PARTIAL CORRELATION COEFFICIENTS

The role of VDE extends beyond encoding node feature dynamics; it also encompasses the critical task of reconstructing the underlying graph structure based on this encoding. To begin, we collect the obtained $\tilde{\mathbf{z}}_i^t$ for all nodes, forming $M$ sets of features denoted as $\mathbf{Z} = \{\mathbf{Z}_{[m]}, 1 \le m \le M\}$, where

$\mathbf{Z}_{[m]} \in \mathbb{R}^{(T-\tau) \times n}$. For each $\mathbf{Z}_{[m]}$, we employ Eqn. 4 to compute a matrix of partial correlations $\mathbf{R}$, where the $(i,j)$-th cell of the matrix denotes the partial correlation of $x_i$ with $x_j$ given a random vector $X_S$, $r_{ij|S}$. Subsequently, the obtained $\mathbf{R}$ is utilized to calculate PCOR, including statistics $t_{ij|S}$ and p-values $p_{ij|S}$, as detailed in (Weatherburn, 1949; Sheskin, 2003):

$$p_{ij|S} = 2\Phi_t\Big(-|t_{ij|S}|, H-2-n\Big), \quad \text{where } t_{ij|S} = r_{ij|S}\sqrt{\frac{H-2-n}{1-r_{ij|S}^2}}. \tag{13}$$

$H$ is the sample size ($= T - \tau$ for a $\mathbf{Z}_m$), $n$ is the number of nodes in the graph. $\Phi_t(\cdot)$ signifies the cumulative density function of a Student's $t$ distribution with degrees of freedom $H - 2 - n$. The resultant structure of the graph is represented as $\mathbf{P}$, with each $(i,j)$-th cell equals to $p_{ij|S}$. Across $M$ trajectories, a collection of $M$ $\mathbf{P}$ matrices is obtained. However, in this symmetric configuration, the derived structure cannot capture the intricacies of directed graphs, which are characterized by unsymmetrical structures. To address this, we propose a novel algorithm that leverages temporal information to infer directed edges.

**From temporal order to directed edges.** In the real-world dynamics of systems, events typically occur in chronological order. Building upon this observation, we utilize temporal information to infer the structure of a directed graph. We reorganize the $M$ sets of features $\mathbf{Z}$ as $\mathbf{Z} = \{\mathbf{Z}_{[m]}, 1 \leq m \leq M\}$, with each $\mathbf{Z}_{[m]} = \{\mathbf{Z}_{[m],i} \in \mathbb{R}^{(T-\tau-1) \times n}, 1 \leq i \leq n\}$. Each row of $\{\mathbf{Z}_{[m],i}\}$ comprises $[\mathbf{z}_0^{t-1}, \mathbf{z}_1^{t-1}, ..., \mathbf{z}_i^{t}, \mathbf{z}_{n-1}^{t-1}]$, combining the dynamics encoding of time step $t$ for node $i$ with the dynamics encoding of time step $t-1$ for the remaining nodes. Moreover, following Eqn. 13, we calculate p-values for each $\{\mathbf{Z}_{[m],i}\}$. This results in a matrix of p-values, denoted as $\mathbf{P}_i$, estimating the directed edges from other nodes to node $i$. We then compose the overall $\mathbf{P}$ as follows: $\mathbf{P}(:, i) = \mathbf{P}_i(:, i)$, yielding an unsymmetrical matrix that effectively estimates the structure of a directed graph. SIDEC adopts this pipeline to reconstruct the structure of directed graphs, i.e., the adjacency matrix $\mathbf{A} = \mathbf{P}$. For implementation details, please refer to Appendix A.7.

**Advantages.** Employing PCOR for structural inference offers several distinct advantages:

1. **Enhanced accuracy:** Benchmark assessments for structural inference have demonstrated that PCOR exhibit superior accuracy (Anonymous, 2023). The utilization of these coefficients ensures high precision in the reconstructed structure.
2. **Dimensionality reduction:** VDE initially reduces the dimensionality of the trajectories, broadening the applicability of PCOR to multi-dimensional trajectories.
3. **Directed graphs:** Incorporating temporal information allows the algorithm to infer directed graph structures, overcoming the limitation of symmetric structures.

## 5 EXPERIMENTS

In our evaluation, we subjected SIDEC to rigorous testing across a diverse range of 20 datasets. These datasets encompassed trajectories featuring both multi-dimensional and one-dimensional features. Our evaluation included robustness testing against trajectories with added Gaussian noise and an in-depth ablation study assessing the influence of time window selection.

### 5.1 GENERAL SETTINGS

**Datasets.** The benchmark we employed for evaluation, StructInfer (Anonymous, 2023), provided a comprehensive set of datasets specifically tailored for structural inference tasks. We selected datasets related to "Vascular Networks", which involved both Springs and NetSims dynamical simulations. These datasets comprised trajectories with varying numbers of nodes, denoted as $n \in \{15, 30, 50, 100, 150, 200, 250\}$. Each dataset was identified based on the type of dynamical simulation and the number of nodes. For instance, "VN_SP_15" denotes trajectories generated using Springs simulation with 15 nodes, while "VN_NS_30" indicates trajectories generated using Net-Sims simulation with 30 nodes. Notably, Springs-generated trajectories possessed four-dimensional features at each time step, whereas NetSims-generated trajectories featured one-dimensional features. This diverse dataset selection allowed us to assess SIDEC's performance across various feature types and its capacity to conduct structural inference on larger graphs. Our data split followed the predefined training, validation, and test sets within the StructInfer benchmark.

Table 1: Average AUROC Results (%) of SIDEC and baselines on StructInfer datasets.

| Methods | VN_SP_15 | VN_SP_30 | VN_SP_50 | VN_SP_100 | VN_SP_150 | VN_SP_200 | VN_SP_250 |
|---|---|---|---|---|---|---|---|
| NRI | $94.5_{\pm 0.01}$ | $95.1_{\pm 0.01}$ | $94.6_{\pm 0.02}$ | $89.2_{\pm 0.02}$ | OOM | OOM | OOM |
| MPM | $96.6_{\pm 0.01}$ | $89.7_{\pm 0.04}$ | $85.1_{\pm 0.02}$ | $84.6_{\pm 0.03}$ | OOM | OOM | OOM |
| ACD | $94.3_{\pm 0.01}$ | $93.7_{\pm 0.01}$ | $87.5_{\pm 0.03}$ | $90.5_{\pm 0.03}$ | OOM | OOM | OOM |
| iSIDG | $96.6_{\pm 0.02}$ | $95.6_{\pm 0.01}$ | $95.7_{\pm 0.02}$ | $85.1_{\pm 0.02}$ | OOM | OOM | OOM |
| RCSI | $97.0_{\pm 0.01}$ | $95.3_{\pm 0.01}$ | $94.5_{\pm 0.02}$ | $90.7_{\pm 0.03}$ | OOM | OOM | OOM |
| SIDEC | $\mathbf{97.6}_{\pm 0.01}$ | $\mathbf{97.0}_{\pm 0.02}$ | $\mathbf{96.5}_{\pm 0.03}$ | $\mathbf{95.7}_{\pm 0.02}$ | $\mathbf{95.3}_{\pm 0.02}$ | $\mathbf{95.5}_{\pm 0.02}$ | $\mathbf{95.0}_{\pm 0.03}$ |

| Methods | VN_NS_15 | VN_NS_30 | VN_NS_50 | VN_NS_100 | VN_NS_150 | VN_NS_200 | VN_NS_250 |
|---|---|---|---|---|---|---|---|
| NRI | $90.3_{\pm 0.01}$ | $74.6_{\pm 0.04}$ | $69.7_{\pm 0.03}$ | $68.8_{\pm 0.02}$ | $60.5_{\pm 0.04}$ | OOM | OOM |
| MPM | $91.2_{\pm 0.01}$ | $83.4_{\pm 0.03}$ | $72.7_{\pm 0.04}$ | $70.3_{\pm 0.03}$ | $63.5_{\pm 0.03}$ | OOM | OOM |
| ACD | $80.3_{\pm 0.02}$ | $65.4_{\pm 0.06}$ | $69.1_{\pm 0.03}$ | $68.7_{\pm 0.03}$ | $62.8_{\pm 0.03}$ | OOM | OOM |
| iSIDG | $91.2_{\pm 0.02}$ | $78.1_{\pm 0.06}$ | $73.7_{\pm 0.02}$ | $68.8_{\pm 0.02}$ | $64.1_{\pm 0.05}$ | OOM | OOM |
| RCSI | $91.5_{\pm 0.02}$ | $82.3_{\pm 0.04}$ | $74.1_{\pm 0.02}$ | $70.3_{\pm 0.03}$ | $64.5_{\pm 0.03}$ | OOM | OOM |
| SIDEC | $\mathbf{96.1}_{\pm 0.03}$ | $\mathbf{97.9}_{\pm 0.01}$ | $\mathbf{98.5}_{\pm 0.02}$ | $\mathbf{98.5}_{\pm 0.01}$ | $\mathbf{98.1}_{\pm 0.03}$ | $\mathbf{98.5}_{\pm 0.02}$ | $\mathbf{98.7}_{\pm 0.03}$ |

In addition to the StructInfer datasets, similar to previous works (Wang & Pang, 2022; Wang et al., 2023a) we conducted evaluations on six directed synthetic biological networks (Pratapa et al., 2020) with the names Linear (LI), Linear Long (LL), Cycle (CY), Bifurcating (BF), Trifurcating (TF), and Bifurcating Converging (BF-CV). These networks represented critical components involved in the differentiation and development of cells (Saelens et al., 2019). To simulate these networks, we used BoolODE (Pratapa et al., 2020), recording trajectories consisting of 49 time steps. Subsequently, we performed subsampling of time steps on these raw trajectories. Trajectories were randomly divided into training, validation, and test sets in a ratio of 8:2:2. In this case, the features at each node represented one-dimensional mRNA expression levels.

**Baselines and metrics.** We compare SIDEC with the following state-of-the-art VAE-based methods for structural inference on directed graphs:

- NRI (Kipf et al., 2018): a VAE-based model for unsupervised relational inference.
- MPM (Chen et al., 2021): an NRI-based method with a relation interaction mechanism and a spatio-temporal message passing mechanism.
- ACD (Löwe et al., 2022): a model that leverages shared dynamics to infer causal relations.
- iSIDG (Wang & Pang, 2022): a VAE-based model that iteratively updates the adjacency matrix to be fed to the encoder with direction information.
- RCSI (Wang et al., 2023a): a reservoir computing network is integrated with a VAE for more efficient structural inference.

To evaluate and compare these methods, we utilized the area under the receiver operating characteristic (AUROC) metric to assess the inferred adjacency matrix against ground truth data. For trajectories from StructInfer, every group of them is the results from three different graphs. For instance, "VN_SP_15" has three different graphs but with the same number of nodes. Following the instructions of the benchmark, we thus run methods on each graph three times, with one extra run on the one with the lowest performance, and then report the average AUROC results for all ten runs. Trajectories generated by synthetic biological networks underwent ten runs with varying random seeds, with the average AUROC results reported.

**Experimental settings.** All experiments were conducted on a single NVIDIA Tesla V100 SXM2 32G graphics card, paired with two Xeon Gold 6132 @ 2.6GHz CPUs. The deep learning methods were trained for a maximum of 600 epochs, with batch sizes, learning rates, and hyperparameters configured according to their respective original implementations. Further details are available in Appendix C, while Appendices D.1-D.2 provide more experimental results and comparison with other structural inference methods.

## 5.2 EXPERIMENTAL RESULTS

In this section, we delve into a comprehensive analysis of the performance of all the methods applied to both VN trajectories from StructInfer and synthetic biological networks. The experimental outcomes of our proposed model and the baseline methods are concisely summarized in Tables 1 and 2. Within Table 1, "OOM" designates "out-of-memory," indicating instances where certain methods couldn't be executed due to memory limitations.

Table 2: Average AUROC Results (%) of SIDEC and baselines on synthetic biological networks.

| Methods | LI | LL | CY | BF | TF | BF-CV |
|---|---|---|---|---|---|---|
| NRI | $70.5\pm_{0.03}$ | $75.0\pm_{0.02}$ | $64.5\pm_{0.03}$ | $59.1\pm_{0.03}$ | $55.1\pm_{0.02}$ | $59.2\pm_{0.04}$ |
| MPM | $75.0\pm_{0.02}$ | $79.2\pm_{0.03}$ | $79.0\pm_{0.03}$ | $63.5\pm_{0.02}$ | $58.4\pm_{0.03}$ | $64.4\pm_{0.03}$ |
| ACD | $65.0\pm_{0.03}$ | $68.4\pm_{0.02}$ | $62.9\pm_{0.02}$ | $59.8\pm_{0.03}$ | $57.2\pm_{0.03}$ | $55.8\pm_{0.03}$ |
| iSIDG | $86.2\pm_{0.02}$ | $88.1\pm_{0.02}$ | $79.5\pm_{0.02}$ | $68.3\pm_{0.02}$ | $60.2\pm_{0.03}$ | $70.7\pm_{0.03}$ |
| RCSI | $88.5\pm_{0.03}$ | $91.0\pm_{0.03}$ | $81.0\pm_{0.02}$ | $72.2\pm_{0.03}$ | $65.0\pm_{0.02}$ | $73.7\pm_{0.02}$ |
| **SIDEC** | $\mathbf{89.0}\pm_{\mathbf{0.02}}$ | $\mathbf{92.3}\pm_{\mathbf{0.03}}$ | $\mathbf{82.4}\pm_{\mathbf{0.02}}$ | $\mathbf{75.8}\pm_{\mathbf{0.03}}$ | $\mathbf{68.5}\pm_{\mathbf{0.03}}$ | $\mathbf{78.5}\pm_{\mathbf{0.02}}$ |

Table analysis shows clear trends. When confronted with trajectories featuring multi-dimensional feature like VN_SP, SIDEC outperforms VAE-based methods significantly in graphs with over 100 nodes, demonstrating remarkable scalability and proficiency in structure reconstruction, a trend also evident in VN_NS trajectories. Contrasting with baseline sensitivity to node count changes, SIDEC consistently excels in reconstruction accuracy. Furthermore, in VN_NS experiments, while baselines appear sensitive to changes in graph node counts, SIDEC maintains consistently high performance in terms of structure reconstruction accuracy. This pattern of superiority is further affirmed by experiments on six synthetic biological networks, with SIDEC's exceptional performance attributed to these key factors:

1. **Utilization of VDE for dynamics encoding:** Unlike conventional VAE-based baseline models, SIDEC leverages a VDE to encode the dynamics of individual nodes into latent variables. This approach operates specifically on node features, avoiding the need to process the entire graph. This feature imparts exceptional scalability, especially for large graphs. Moreover, it stands out by not imposing a uniform prior on the inferred structure.

2. **Synergy between PCOR and VDE:** As evident from the StructInfer benchmark, the calculation of PCOR, when used in isolation, exhibits remarkable accuracy in reconstructing graph structures. However, its applicability is constrained, as it cannot handle multi-dimensional features or infer structures in directed graphs. In our proposed methods, VDE plays a dual role: it reduces feature dimensionality and encodes dynamics into the latent variable, which is subsequently utilized in the calculation of PCOR. This symbiotic relationship harnesses the strengths of both PCOR and VDE, resulting in significantly enhanced performance.

3. **Introducing directionality through temporal ordering:** SIDEC introduces directional edges by considering temporal ordering. This innovation substantially broadens the utility of PCOR for structural inference in directed graphs. As demonstrated in the tables, this extension seamlessly restores the accuracy and scalability of PCOR.

4. **Versatility in edge priors:** SIDEC does not rely on a uniform prior for edges, making them exceptionally versatile. They can be applied across a broader spectrum of graph types. The VN datasets, characterized by diverse properties, resist summarization via a uniform prior. This is precisely where SIDEC outperforms VAE-based methods.

## 5.3 ROBUSTNESS TESTS

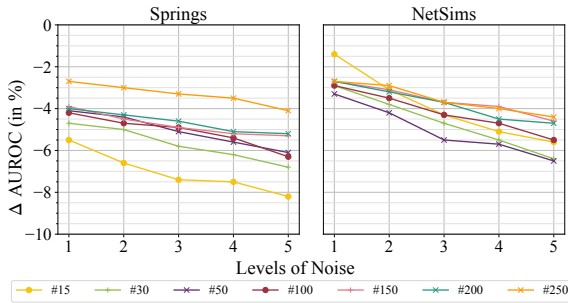

Figure 2: Performance differences (in %) of SIDEC on VN trajectories with Gaussian noise of different levels.

In this section, we delve into the performance degradation of SIDEC when applied to VN trajectories from StructInfer, which have been subjected to various levels of added Gaussian noise. We utilize two sets of datasets: VN_SP and VN_NS trajectories, each intentionally perturbed with five distinct levels of Gaussian noise from the StructInfer dataset. To quantify the performance differences, we calculate $\Delta\text{AUROC} = \text{AUROC}_{NX} - \text{AUROC}_{N0}$, where $\text{AUROC}_{N0}$ signifies the average AUROC results for pristine, noise-free trajectories, and $\text{AUROC}_{NX}$ represents the average AUROC values obtained under $X$ level of added Gaussian noise, $X \in \{1, 2, 3, 4, 5\}$. These results are visually presented in Fig. 2. As evident from both figures within Fig. 2, the inclusion of noise in trajectories indeed exerts a negative influence on the performance of SIDEC. This perfor-

mance degradation is linearly correlated with the intensity of noise. However, it is noteworthy that despite the noise, the performance drops remain below 10%. This observation underscores SIDEC's robustness against added Gaussian noise, ranging from moderate to high. Interestingly, the left side of Fig. 2 reveals that the performance drops for larger graphs are consistently smaller than those observed for smaller graphs. For instance, the line representing the performance drop on VN_SP_30 consistently surpasses that of VN_SP_15, and a similar trend is noted for other line pairs. In contrast, the right side of Fig. 2 presents a less distinct pattern for VN_NS trajectories, with lines exhibiting greater overlap. This intriguing observation suggests that SIDEC excels at extracting and encoding rich information from multi-dimensional features into the latent variable. When applied to larger graphs, this results in a more substantial amount of information encoded within the latent variable, enabling SIDEC to effectively mitigate the adverse effects of noisy features.

## 5.4 Ablation Study on the Length of Time Window

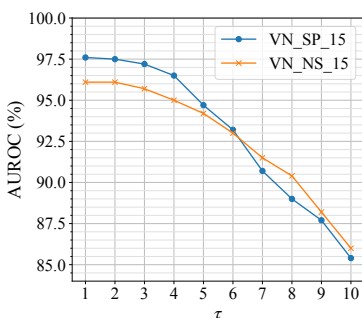

Figure 3: Average AUROC values (in %) of SIDEC-T on noise-free VN_SP_15 and VN_NS_15 trajectories with different values of $\tau$.

In this section, we delve into the critical aspect of selecting the appropriate length for the time window, denoted as $\tau$, which has the potential to significantly impact the performance of SIDEC when applied to VN trajectories sourced from Struct-Infer. As mentioned in Section 4.1, $\tau$ is chosen based on the dynamics between $v^t$ and $v^{t+\tau}$ in a manner that ensures the system's dynamics exhibit Markovian behavior. However, due to the continuous evolution of dynamical systems and the inherent discrete nature of the obtained trajectories, our assessment of the system's Markovian properties is an approximation. To pinpoint the most suitable value for the time window $\tau$, we conduct an ablation study using noise-free VN_SP_15 and VN_NS_15 trajectories. Our test set for $\tau$ encompasses values drawn from the set $\{1, 2, 3, 4, 5, 6, 7, 8, 9, 10\}$, and the results are averaged over ten independent runs, presented in Fig. 3. As evident from the figure, when $\tau$ increases, the performance of SIDEC on both VN_SP_15 and VN_NS_15 gradually deteriorates, with more pronounced performance drops observed at larger $\tau$ values. This phenomenon stems from the fact that selecting an excessively large time window disrupts the underlying mechanism of VDE. When too many time steps are skipped, the node feature dynamics become excessively chaotic, rendering VDE less effective at capturing and encoding these dynamics. Consequently, based on the insights derived from this ablation study, we opt for $\tau = 1$ as the most suitable choice.

## 6 Conclusion

In this paper, we introduced an innovative structural inference method, employing VDE in tandem with partial correlation coefficients. VDE serves a dual purpose: encoding node dynamics into latent variables and extracting essential information from node features to reduce dimensionality. These latent variables enable precise graph reconstruction via partial correlation coefficients, accommodating trajectories with one-dimensional and multi-dimensional features. Notably, our approach sidesteps the need for a comprehensive graph view, granting it exceptional scalability for large graphs, a long-standing challenge. Through temporal reordering of latent variables, our method is able to reconstruct directed graph structures. Experimental results validate its efficacy across diverse datasets and large-scale graphs, showcasing robustness against Gaussian noise.

Currently, due to the formidable challenges involved in acquiring real-world data, our experimental validation is confined to synthetic data. A promising avenue for future research lies in the collection and curation of real-world datasets tailored for structural inference tasks. Furthermore, inspired by recent work (Wang et al., 2023b), the exploration of individualized local receptive fields for each node presents another intriguing avenue for future investigation. Such local biases, when incorporated during structural inference, hold the potential to uncover deeper insights into complex real-world networks. Besides that, it would also be interesting to develop a structural inference method that can deal with the incomplete view of nodes, where some nodes are unobservable. These promising research directions pave the way for advancements in structural inference.

## Reproducibility Statement

To ensure both the reproducibility and comprehensiveness of this paper, we have included an appendix containing detailed implementation information. Additionally, the code can be found at `https://github.com/wang422003/SIDEC_torch`. Appendix A.1 presents the pseudo-code outlining the PCOR method used in this work, while Appendix A.6 offers the pseudo-code for the complete SIDEC pipeline. Furthermore, Appendix A.7 furnishes a comprehensive list of instructions on how to construct SIDEC from scratch. For a thorough understanding of the datasets employed in our research, please refer to Appendix B, where we provide detailed descriptions of their characteristics and sources. With these comprehensive resources, we have taken diligent steps to ensure the reproducibility of our work.

## Acknowledgments

Author Jun Pang acknowledges financial support from the Institute for Advanced Studies of the University of Luxembourg through an Audacity Grant (AUDACITY-2021). The experiments presented in this paper were carried out using the HPC facilities of the University of Luxembourg (Varrette et al., 2022) (see `hpc.uni.lu`).

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

# A   MORE DETAILS ABOUT SIDEC

## A.1   PARTIAL CORRELATIONS

In this section, we provide a derivation of the partial correlation mentioned in Section 3. The derivation strictly follows the description provided by Kim (2015). Consider the random vector $X = (x_1, x_2, ..., x_i, ..., x_n)'$ where $|X| = n$. We denote the variance of a variable random $x_i$ and the covariance between two random variables $x_i$ and $x_j$ as $Var_i (= var(x_i)$ and $Cov_{ij} (= cov(x_i, x_j)$, respectively. The variance-covariance matrices of random vectors $X$ and $X_S$ ($S \subset \{1, 2, ..., n\}$ and $|S| < n$) are denoted by $C_X$ and $C_S$, respectively, where $X_S$ is a random sub-vector of the random vector $X$. The correlation between two random variables $x_i$ and $x_j$ is denoted by $r_{ij} = \frac{Cov_{ij}}{\sqrt{Var_i}\sqrt{Var_j}} (= corr(x_i, x_j))$. Thus, the partial correlation $r_{ij|k}$ of $x_i$ with $x_j$ given $x_k$ is:

$$r_{ij|k} = \frac{r_{ij} - r_{ik}r_{jk}}{\sqrt{1 - r_{ik}^2}\sqrt{1 - r_{jk}^2}}, \tag{14}$$

Interestingly, we can easily obtain another version of the partial correlation as:

$$r_{ij|k} = corr(resid(i|k), resid(j|k))$$

$$= \frac{Cov_{ij} - Cov_{ik}Var_k^{-1}Cov_{kj}}{\sqrt{Var_i - Cov_{ik}Var_k^{-1}Cov_{ki}}\sqrt{Var_j - Cov_{jk}Var_k^{-1}Cov_{kj}}}, \tag{15}$$

where $resid(i|k) = x_i - \hat{x}_i(x_k)$ and $\hat{x}_i(x_k) = Cor_{ik}Var_k^{-1}x_k$. This form concurrence the definition of partial correlation in (Whittaker, 1990).

With this equivalence, we can introduce the calculation of higher-order partial correlations proposed by Whittaker (1990). To do this, we need to consider the inverse variance-covariance matrix of $X$, $D_X = C_X^{-1}$. We denote $d_{ij}$ and $c_{ij}$ are the $(i, j)$-th cell of the matrices $D_X$ and $C_X$, respectively. Then following (Whittaker, 1990) the partial correlation of $x_i$ with $x_j$ given a random vector $X_S$ is:

$$r_{ij|S} = -\frac{d_{ij}}{\sqrt{d_{ii}}\sqrt{d_{jj}}}, \tag{16}$$

where $X_S$ is the random sub-vector of $X$ after removing the random variables $x_i$ and $x_j$ and its size is $|S| = |X| - 2$. Through inversion of $D_X$, all partial correlations between $x_i$ and $x_j$ given all other variables $X_S$ are simultaneously calculated. This formulation is in practice a more efficient calculation than the recursion formula proposed by Anderson (1958). We suggest interested readers to refer to the literature for more details. Thus the introduction of Eqn. 4 in Section 3 is validated.

## A.2   AUTOCORRELATION LOSS

As elaborated in Section 4.2, the autocorrelation loss term $\mathcal{L}_{\text{autoc}}$, outlined in Equation 11, plays a pivotal role in enhancing the network's capacity to capture the extended timescale kinetics within time-series data (Hernández et al., 2018). While the minimization of the reconstruction loss holds the potential to recover these intricate dynamical processes, it alone proves insufficient. To ensure robust model convergence, the original VDE paper (Hernández et al., 2018) drew inspiration from the variational approach to conformational dynamics (VAC) (Noé & Nuske, 2013). In this context, the authors introduced the autocorrelation loss function. This innovation allowed for the estimation of autocorrelations using multiple short simulations distributed across the state space, eliminating the necessity for direct simulation of slow processes within a single, lengthy trajectory. This feature confers a critical advantage in addressing the sampling challenges that frequently plague dynamical systems investigations. In line with this approach, we incorporate the autocorrelation loss into SIDEC. For a more comprehensive understanding of the derivation of Equation 11, we refer to the insights offered by Noé & Nuske (2013). Following the insights, here we provide more details about the derivation of Eqn. 11.

**Basics.**   Let $\Omega$ be a state space, and use $\mathbf{x}$, $\mathbf{y}$ to denote points in this state space. We consider a Markov process $\mathbf{mp}_t \in \Omega$ which is stationary and ergodic with respect to its unique stationary (invariant) distribution $\mu(\mathbf{x}) \equiv p(\mathbf{mp}_t = \mathbf{x})$ $\forall t$. The dynamics of the process $\mathbf{mp}_t$ are characterized

by the transition density

$$p(\mathbf{x}, \mathbf{y}; \tau) = p(\mathbf{mp}_{t+\tau} = \mathbf{y} | \mathbf{mp}_t = \mathbf{x}), \tag{17}$$

which we assume to be independent of the time $t$. The correlation density, i.e., the probability density of finding the process at points $\mathbf{x}$ and $\mathbf{y}$ at a time spacing of $\tau$, is then defined by

$$C(\mathbf{x}, \mathbf{y}; \tau) = \mu(\mathbf{x}) p(\mathbf{x}, \mathbf{y}; \tau) = p(\mathbf{mp}_{t+\tau} = \mathbf{y}, \mathbf{mp}_t = \mathbf{x}). \tag{18}$$

We further assume $\mathbf{mp}_t$ to be reversible with respect to its stationary distribution, i.e.:

$$\mu(\mathbf{x}) p(\mathbf{x}, \mathbf{y}; \tau) = \mu(\mathbf{y}) p(\mathbf{x}, \mathbf{y}; \tau) \tag{19}$$

$$C(\mathbf{x}, \mathbf{y}; \tau) = C(\mathbf{y}, \mathbf{y}; \tau) \tag{20}$$

Reversibility is not strictly necessary but tremendously simplifies the forthcoming expressions and their interpretation (Sarich et al., 2010). If at time $t = 0$, the process is distributed according to a probability distribution $\rho_0$, the corresponding distribution at time $\tau$ is given by:

$$\rho_\tau(\mathbf{y}) = \int_\Omega d\mathbf{x} \rho_0(\mathbf{x}) p(\mathbf{x}, \mathbf{y}; \tau) =: \mathcal{P}(\tau) \rho_0. \tag{21}$$

The time evolution of probability densities can be seen as the action of a linear operator $\mathcal{P}(\tau)$, called the propagator of the process. This is a well-defined operator on the Hilbert space $L^2_{\mu^{-1}}(\Omega)$ of functions which are square-integrable with respect to the weight function $\mu^{-1}$. The scalar-product on this space is given by

$$\langle u | v \rangle_{\mu^{-1}} = \int_\Omega d\mathbf{x} u(\mathbf{x}) v(\mathbf{x}) \mu^{-1}(\mathbf{x}). \tag{22}$$

If we assume the transition density to be a smooth and bounded function of $\mathbf{x}$ and $\mathbf{y}$, the propagator can be shown to be bounded, with operator norm less or equal to one. Since the stationary of $\mu$ implies $\mathcal{P}(\tau)\mu = \mu$, we have $\|\mathcal{P}(\tau)\| = 1$. Reversibility allows us to show that the propagator is self-adjoint and compact. Furthermore, using the definition of the transition density, we can show that $\mathcal{P}(\tau)$ satisfies a Chapman-Kolmogorov equation: For times $\tau_1, \tau_2 \geq 0$, we have:

$$\mathcal{P}(\tau_1 + \tau_2) = \mathcal{P}(\tau_1) \mathcal{P}(\tau_2). \tag{23}$$

**Spectral decomposition.** It follows from the above arguments that $\mathcal{P}(\tau)$ possesses a sequence of real eigenvalues $\lambda_i(\tau)$ with $|\lambda_i(\tau)| \leq 1$ and $|\lambda_i(\tau)| \to 0$. Each of these eigenvalues corresponds to an eigenfunction $l_i \in L^2_{\mu^{-1}}(\Omega)$. The functions $l_i$ form an orthonormal basis of the Hilbert space $L^2_{\mu^{-1}}(\Omega)$. Clearly, $\lambda_1(\tau) = 1$ is an eigenvalue with eigenfunction $l_1 = \mu$. In many applications, we can assume that $\lambda_1(\tau)$ is non-degenerate and $-1$ is not an eigenvalue. Additionally, there usually is a number $m$ of positive eigenvalues

$$1 = \lambda_1(\tau) > \lambda_2(\tau) > ... > \lambda_m(\tau), \tag{24}$$

which are separated from the remaining spectrum. Because of the Chapman-Kolmogorov equation, each eigenvalue $\lambda_i(\tau)$ decays exponentially in time, i.e. we have

$$\lambda_i(\tau) = \exp(-\kappa_i \tau), \tag{25}$$

for some rate $\kappa_i \geq 0$. Clearly, $\kappa_1 = 0$, $\kappa_2, ..., \kappa_m$ are close to zero, and all remaining rates are significantly larger than zero. If we now expand a function $u \in L^2_{\mu^{-1}}(\Omega)$ in terms of the functions $l_i$, i.e.

$$u = \sum_{i=1}^{\infty} \langle u | l_i \rangle_{\mu^{-1}} l_i, \tag{26}$$

we can decompose the action of the operator $\mathcal{P}(\tau)$ into its action on each of the basis functions:

$$
\begin{aligned}
\mathcal{P}(\tau) u &= \sum_{i=1}^{\infty} \langle u | l_i \rangle_{\mu^{-1}} \mathcal{P}(\tau) l_i \\
&= \sum_{i=1}^{\infty} \lambda_i(\tau) \langle u | l_i \rangle_{\mu^{-1}} l_i \\
&= \sum_{i=1}^{\infty} \exp\left(-\kappa_i \tau\right) \langle u | l_i \rangle_{\mu^{-1}} l_i.
\end{aligned}
\tag{27}
$$

For time windows $\tau \gg \frac{1}{\kappa_{m+1}}$, all except the first $m$ terms in the above sum have become very small (Sarich et al., 2010), and to a good approximation we have

$$\mathcal{P}(\tau)u \approx \sum_{i=1}^{m} \exp\left(-\kappa_i\tau\right)\langle u|l_i\rangle_{\mu^{-1}} l_i. \tag{28}$$

Knowledge of the dominant eigenfunctions and eigenvalues is therefore most helpful to the understanding of the process.

**Rayleigh variational principle.** In nontrivial dynamical systems neither the correlation densities $p(\mathbf{x}, \mathbf{y}; \tau)$ and $C(\mathbf{x}, \mathbf{y}; \tau)$ nor the eigenvalues $\lambda_i(\tau)$ and eigenfunctions $l_i$ are analytically available. Noé & Nuske (2013) provides a variational principle based on which these quantities can be estimated from simulation data generated by the dynamical process $\mathbf{mp}_t$. For this, the formalism introduced above is used to formulate the Rayleigh variational principle used in quantum mechanics (Szabo & Ostlund, 2012) for Markov processes. Let $f$ be a real-valued function of state, $f = f(\mathbf{x}) : \Omega \to \mathbb{R}$. Its autocorrelation with respect to the stochastic process $\mathbf{mp}_t$ is given by

$$\text{autoc}(f; \tau) = \mathbb{E}[f(\mathbf{mp}_0)f(\mathbf{mp}_\tau)] = \int_{\mathbf{x}}\int_{\mathbf{y}} d\mathbf{x}d\mathbf{y} f(\mathbf{x})C(\mathbf{x}, \mathbf{y}; \tau)f(\mathbf{y}) = \langle \mathcal{P}(\tau)\mu f|\mu f\rangle_{\mu^{-1}}. \tag{29}$$

In the Dirac notation often used in physical literature, integrals such as the one above may be abbreviated by $\mathbb{E}[f(\mathbf{x}_0)f(\mathbf{x}_\tau)] = \langle \mu f|\mathcal{P}(\tau)|\mu f\rangle$. We may easily get that the autocorrelation function of a weighted eigenfunction $r_k = \mu^{-1}l_k$ is:

$$\begin{aligned}
\text{autoc}(r_k; \tau) &= \mathbb{E}[r_k(\mathbf{mp}_0)r_k(\mathbf{mp}_\tau)] \\
&= \langle \mathcal{P}(\tau)l_k|l_k\rangle_{\mu^{-1}} \\
&= \lambda_k(\tau)\langle l_k|l_k\rangle_{\mu^{-1}} \\
&= \lambda_k(\tau),
\end{aligned} \tag{30}$$

where $\lambda_k(\tau)$ is the eigenvalue of $r_k$.

Let $\hat{l}_2$ be an approximate model for the second eigenfunction, which is normalized and orthogonal to the true first eigenfunction: $\langle \hat{l}_2, \mu\rangle_{\mu^{-1}} = 0$ and $\langle \hat{l}_2, \hat{l}_2\rangle_{\mu^{-1}} = 1$. Then according to (2.24) in (Noé & Nuske, 2013), we find for $\hat{r}_2 = \mu^{-1}\hat{l}_2$:

$$\text{autoc}(\hat{r}_2; \tau) = \mathbb{E}[\hat{r}_2(\mathbf{mp}_0)\hat{r}_2(\mathbf{mp}_\tau)] \leq \lambda_2(\tau). \tag{31}$$

Similarly, let $\hat{l}_k$ be an approximate model for the $k$'th eigenfunction, with the normalization and orthogonality constraints: $\langle \hat{l}_k, l_i\rangle_{\mu^{-1}} = 0, \forall i < k$ and $\langle \hat{l}_k, \hat{l}_k\rangle_{\mu^{-1}} = 1$, then

$$\text{autoc}(\hat{r}_k; \tau) = \mathbb{E}[\hat{r}_k(\mathbf{mp}_0)\hat{r}_k(\mathbf{mp}_\tau)] \leq \lambda_k(\tau). \tag{32}$$

An important insight at this point is that a variational principle of conformation dynamics can be formulated in terms of correlation functions. In contrast to quantum mechanics or other fields where the variational principle has been successfully employed, no closed-form expression of the operator $\mathcal{P}(\tau)$ is needed. The ability to express the variational principle in terms of correlation functions with respect to $\mathcal{P}(\tau)$ means that the eigenvalues to be maximized can be directly estimated from data. If statistically sufficient realizations of $\mathbf{mp}_t$ are available, then the autocorrelation function can be estimated via

$$\text{autoc}(\hat{r}_k; \tau) = \mathbb{E}(\hat{r}_k(\mathbf{mp}_0)\hat{r}_k(\mathbf{mp}_\tau)) \approx \frac{1}{N}\sum \hat{r}_k(\mathbf{mp}_0)\hat{r}_k(\mathbf{mp}_\tau), \tag{33}$$

where $N$ is the number of simulated time windows of length $\tau$.

**For Structural Inference.** This section provides the contextual foundation for the formulations presented above. The variational principle posits that, in the limit of an infinite dataset, no process within the data can be identified that is slower than the true underlying process. However, our aim is to encode the dynamics of individual nodes into latent variables, capturing the true essence of the dynamical system's behavior. Therefore, we must establish a metric to gauge the suitability of the current VDE in modeling the true process. This metric ensures that the latent variable $\mathbf{z}_t$ effectively encapsulates the dynamics that faithfully represent the entire process.

As expressed in Equation 28, acquiring knowledge of the dominant eigenfunctions $l_i$ and their corresponding eigenvalues $\lambda_i$ proves most valuable for approximating the underlying process. Consequently, we turn to the Rayleigh variational principle to estimate these critical quantities from data

---

**Algorithm 1** From temporal order to directed edges.

1: **Input:** $M$ sets of latent variables as $\mathbf{Z} = \{\mathbf{Z}_{[m]}, 1 \leq m \leq M\}$, where $\mathbf{Z}_{[m]} \in \mathbb{R}^{(T-\tau) \times n}$,
2: **Output:** A set of matrices of p-values representing the relation between $n$ nodes: $\{\mathbf{P}\}$,
3: **for** node $i$ in total $n$ nodes **do**
4: $\quad$ $\mathbf{Z}_{[m],i} = [\mathbf{z}_0^{t-1}, \mathbf{z}_1^{t-1}, ..., \mathbf{z}_i^t, \mathbf{z}_{n-1}^{t-1}]$,
5: **end for**
6: Constitute set $\mathbf{Z} = \{\mathbf{Z}_{[m]}, 1 \leq m \leq M\}$, with each $\mathbf{Z}_{[m]} = \{\mathbf{Z}_{[m],i} \in \mathbb{R}^{(T-\tau-1) \times n}, 1 \leq i \leq n\}$,
7: **for** trajectory $m$ in total $M$ trajectories **do**
8: $\quad$ **for** node $i$ in total $n$ nodes **do**
9: $\quad\quad$ Calculate the variance-covariance matrix $C_X$ based on $\mathbf{Z}_{[m],i}$,
10: $\quad\quad$ Calculate $D_X = C_X^{-1}$,
11: $\quad\quad$ **for** node $j$ **do**
12: $\quad\quad\quad$ Calculate partial correlation $r_{ij|S} = -\frac{d_{ij}}{\sqrt{d_{ii}}}\sqrt{d_{jj}}$ (Eqn. 4),
13: $\quad\quad\quad$ Calculate $t_{ij|S} = r_{ij|S}\sqrt{\frac{H-2-n}{1-r_{ij|S}^2}}$,
14: $\quad\quad\quad$ Calculate $p_{ij|S} = 2\Phi_t\Big(-|t_{ij|S}|, H-2-n\Big)$,
15: $\quad\quad$ **end for**
16: $\quad\quad$ Form $\mathbf{P}_{[m],i}$ based on $p_{ij|S}$,
17: $\quad$ **end for**
18: $\quad$ Form $\mathbf{P}_{[m]}(:, i) = \mathbf{P}_{[m],i}(:, i)$ for all node $i$,
19: **end for**
20: Return the set of p-value matrices $\{\mathbf{P}\} = \{\mathbf{P}_{[m]}, 1 \leq m \leq M\}$.

---

generated by the dynamical process, simultaneously striving to maximize the eigenvalues. As outlined in Equation 33, the maximization of eigenvalues aligns with the maximization of the estimated autocorrelation function. Notably, in our specific case characterized by a single linear decomposition of the phase-space limit, only one eigenvalue warrants consideration. This eigenvalue corresponds to the autocorrelation of the latent variable $\mathbf{z}_t$, giving rise to Equation 11 as elucidated in Section 4.2.

### A.3 FROM TEMPORAL ORDER TO DIRECTED EDGES

In this section, we discuss more about the method mentioned in Section 4.3, which is leveraged to help PCOR to reconstruct the structure of directed graphs. PCOR is initially designed to measure the correlations between variables. Thus it cannot provide the directionality of the relation between variables, as the calculations of correlation has no directionality. However, if we take the temporal information into consideration, and would figure out that there actually exists a temporal directed connection in the time series. Suppose we have three nodes, $i$, $j$, and $k$. In the underlying interaction graph, there exists a directed edge from $j$ to $i$, while the node $k$ is disconnected from the rest. We sample the node features within a time period $T$, and obtain $\{V\} = \{V^0, V^1, ..., V^{T-1}\}$ as the trajectory of the system. At time step $t$, we have $V^t = \{v_i^t, v_j^t, v_k^t\}$.

If we just use node features at the same time step for PCOR calculation, it would be impossible to figure out the directed edge from $j$ to $i$, as the partial correlation coefficients $p_{ij|S} = p_{ji|S}$. However, we may recompose the time series which serves as the input for PCOR. Instead of feeding PCOR with $\{V\} = \{V^0, V^1, ..., V^{T-1}\}$, whose element $V^t = \{v_i^t, v_j^t, v_k^t\}$, we rearrange the node features for every time step. For example, if we would like to figure out which other node may have a directed edge to node $i$, we construct a new feature trajectory $\{V_i\} = \{V_i^1, V_i^2, ..., V_i^{T-1}\}$, where at time step $t$, $V_i^t = \{v_i^t, v_j^{t-1}, v_k^{t-1}\}$. In this formulation, every present feature of node $i$ is paired with the feature from the previous time step of other nodes. We then feed $\{V_i\}$ to PCOR to obtain a $\mathbf{P}_i$ representing the measurement of the relation between node $i$ and other nodes. If we detect there is a connection between $j$ and $i$ in the obtained $\mathbf{P}_i$, because the events typically occur in chronological order and with the pairing of node features, the result can only lead to there is a directed edge from $j$ to $i$. For other nodes, we calculate the $\mathbf{P}_j$, $\mathbf{P}_k$ following the above-mentioned steps. After obtaining all coefficient matrices, we constitute the final coefficient matrix for the entire system $\mathbf{P}$ as $\mathbf{P}(:, i) = \mathbf{P}_i(:, i)$. We may replace $v_i^t$ in the above formulation with $z_i^t$, and thus we

---

**Algorithm 2** Overall pipeline of SIDEC.

---

1: **Input:** $M$ trajectories of node features as $\mathbf{V} = \{\mathbf{V}_{[m]}, 1 \leq m \leq M\}$. Each trajectory consists of the features of $n$ nodes spanning from time $[0, T-1]$, which is marked as $v_{[m],i}^t$ for node $i$ at time $t$ in trajectory $m$.
2: **Parameters:** Number of epochs $E$, batch size $B$, time window length $\tau$,
3: **Output:** A set of matrices of p-values representing the relation between $n$ nodes: $\{\mathbf{P}\}$,
4: Flatten all node features from all node trajectories and form a new set $V_{\text{all}}$,
5: Constitute $N_b$ batches from $V_{\text{all}}$ with size $B$,
6: **for** batch $X_b$ in total $N_b$ batches **do**
7:     Train encoder $ENC$ and obtain $\{\mu_i^t, \sigma_i^t\}$ for current batch (Eqn. 6),
8:     Obtain $\{\mathbf{z}\}$ based on reparameterization step (Eqn. 7),
9:     Train decoder $DEC$ and obtain $\{\hat{v}_i^{t+\tau}\}$ for current batch (Eqn. 8),
10:     Calculate reconstruction loss: $\mathcal{L}_{\text{recon}} = \frac{1}{L} \sum \|\hat{v}^{t+\tau} - v^{t+\tau}\|^2 - \frac{1}{2L} \sum (1 + \log \sigma^2 - \sigma^2 - \mu^2)$ (Eqn. 10),
11:     Calculate autocorrelation loss: $\mathcal{L}_{\text{autoc}} = -\frac{\mathbb{E}[(\mu^t - \bar{\mu}^t)(\mu^{t+\tau} - \bar{\mu}^{t+\tau})]}{\mathbf{s}_{\mu^t} \mathbf{s}_{\mu^{t+\tau}}}$ (Eqn. 11),
12:     Calculate loss for VDE: $\mathcal{L}_{\text{VDE}} = \mathcal{L}_{\text{recon}} + \mathcal{L}_{\text{autoc}}$,
13:     Update weights of encoder and decoder,
14: **end for**
15: After training, feed $V_{\text{all}}$ into VDE again, and collect the $\tilde{\mathbf{z}} = \{\mu\}$ for every input,
16: Reshape $\tilde{\mathbf{z}}$ according to the length of the trajectory and the number of nodes, and obtain $\mathbf{Z} = \{\mathbf{Z}_{[m]}, 1 \leq m \leq M\}$, where $\mathbf{Z}_{[m]} \in \mathbb{R}^{(T-\tau) \times n}$,
17: Follow steps in Algorithm 1 to obtain the set of p-value matrices $\{\mathbf{P}\} = \{\mathbf{P}_{[m]}, 1 \leq m \leq M\}$,
18: Return the set of p-value matrices $\{\mathbf{P}\} = \{\mathbf{P}_{[m]}, 1 \leq m \leq M\}$.

---

obtain the procedure mentioned in Section 4.3. We summarize the pipeline for inferring directed edges with PCOR in our work in Algorithm 1.

### A.4    THE LENGTH OF TIME WINDOW

In the context of the paper, when it is mentioned that $\tau$ represents the length of the time window required for the system's dynamics to be considered Markovian, it is referring to the concept of Markovianity in continuous-time Markov processes (Noé & Nuske, 2013). In these processes, the future state of the system is determined solely by its current state and not by how it arrived in that state. This property is essential to simplify the modeling and analysis of dynamical systems.

The time window $\tau$ is crucial because it sets the scale at which the dynamics are examined. For a system to be considered Markovian, the choice of $\tau$ should be such that the memory of past states does not significantly influence the future states beyond this time window. In practical terms, this means that if you know the state of the system at time $\tau$, then this information is sufficient to predict its state at time $t + \tau$ without needing to know its earlier history.

The impact of the time window $\tau$ on the Markovian nature of the system can be understood as:

1. If $\tau$ is too short, the system's evolution may still be influenced by its immediate past, not adhering to the Markovian property.

2. If $\tau$ is appropriately chosen, it allows for the dynamics of the system to 'forget' its past states, thus adhering more closely to the Markovian assumption. This is critical for the effective application of Markov models and for simplifying the analysis of the system's dynamics. This could be the case of analyzing open quantum system which somehow expresses its non-Markovian characteristic.

In these scenarios, a $\tau$ value larger than 1 might be more appropriate. In the case of the StructInfer (Anonymous, 2023) datasets, which were generated with smaller time intervals and subsequently subsampled, a $\tau$ value of 1 suggests that the system behaves Markovian over these intervals. This means that past states do not significantly influence future states beyond this time window.

Exploring how the length of the time window affects various dynamical systems and sampling frequencies in detail will deepen our understanding of the VDE and consequently broaden the application scenario of SIDEC. We plan to delve into this research area in future work.

## A.5   FULL TRAJECTORIES OR JUST ONE?

It is worth investigating the effectiveness of SIDEC with just one trajectory as input and with all trajectories as input. We gathered experimental data to compare the performance of SIDEC with full node trajectories (SIDEC(*Full)) and single-node trajectories. Table 3 presents our findings.

Table 3: Average AUROC Results (%) of SIDEC and SIDEC(*Full) on VN_SP trajectories.

| Methods | VN_SP_15 | VN_SP_30 | VN_SP_50 | VN_SP_100 | VN_SP_150 | VN_SP_200 | VN_SP_250 |
|---|---|---|---|---|---|---|---|
| SIDEC(*Full) | $95.8_{\pm 0.02}$ | $96.2_{\pm 0.03}$ | $96.0_{\pm 0.02}$ | $95.0_{\pm 0.01}$ | $94.6_{\pm 0.03}$ | OOM | OOM |
| SIDEC | $\mathbf{97.6}_{\pm \mathbf{0.01}}$ | $\mathbf{97.0}_{\pm \mathbf{0.02}}$ | $\mathbf{96.5}_{\pm \mathbf{0.03}}$ | $\mathbf{95.7}_{\pm \mathbf{0.02}}$ | $\mathbf{95.3}_{\pm \mathbf{0.02}}$ | $\mathbf{95.5}_{\pm \mathbf{0.02}}$ | $\mathbf{95.0}_{\pm \mathbf{0.03}}$ |

The results indicate that SIDEC(*Full) is consistently outperformed by SIDEC using single-node trajectories. Additionally, using full trajectories leads to scalability issues on graphs with more than 150 nodes. It also complicates the identification of latent variables corresponding to specific nodes. These findings justified our choice to focus on single-node trajectories in the VDE stage for efficiency and efficacy.

## A.6   OVERALL PIPELINE

We summarize the overall pipeline of SIDEC in Algorithm 2.

---

**Algorithm 3** Pseudocode for the encoder in VDE.

---
1: **Input:** features $input$
2: x = mlp1($input$)
3: x = mlp2(x)
4: x = mlp3(x)
5: out = mlp4(x)
6: **Return:** out

---

**Algorithm 4** Pseudocode for the decoder in VDE.

---
1: **Input:** features $input$
2: x = mlp5($input$)
3: x = mlp6(x)
4: x = mlp7(x)
5: out = mlp8(x)
6: **Return:** out

---

## A.7   IMPLEMENTATION

In this section, we provide details about the implementation of the proposed SIDEC. Besides the description in this section, readers may also refer to the code in the following GitHub Repository: `https://github.com/wang422003/SIDEC_torch`.

**Implementation of VDE.**   The basic class of VDE is implemented as the inheritance of VAE Class of "PyTorch-VAE" (Subramanian, 2020), with the following modification:

1. **New encoder and decoder:** We implement the encoder and decoder according to the VDE implementation (Hernández et al., 2018). We enclose the setup of the encoder and decoder as pseudo code and shown in Algorithms 3 and 4. The implementation of MLP in the encoder and decoder is described in Algorithm 5 with parameters shown in Table 4.
2. **Changes in loading data:** We integrated the data loading pipelines from iSIDG implemented by StructInfer (Anonymous, 2023) into the VDE data loading pipeline.
3. **New loss functions:** Besides the reconstruction loss term that is implemented already in the VAE Class, we implemented the pipeline for autocorrelation loss and the summation of both reconstruction loss and autocorrelation loss.
4. **Addition of "predict()" method:** We added a method in the VAE class to run the trained model again and sample the latent variables.

---

**Algorithm 5** An MLP block.

---
1: **Input:** features $input$
2: x = Linear($input$)
3: x = Swish(x)
4: out = DropOut(x)
5: **Return:** out

---

Table 4: Parameters in the Linear layers in MLPs in encoder and decoder.

| BLOCK | Number of Units | Dropout Rates |
|-------|-----------------|---------------|
| MLP1 | 2048 | 0.0 |
| MLP2 | 2048 | 0.3 |
| MLP3 | 2048 | 0.3 |
| MLP4 | 1 | 0.0 |
| MLP5 | 2048 | 0.0 |
| MLP6 | 2048 | 0.3 |
| MLP7 | 2048 | 0.3 |
| MLP8 | Original Dimension $d$ | 0.0 |

**Calculation of Partial Correlation Coefficients.** The main pipeline of the calculation of partial correlation coefficients follows the one provided in `https://gist.github.com/fabianp/9396204419c7b638d38f`, which is a Python implementation of the calculation of partial correlations. But we need to calculate the p-values, thus we extend this function with the following modification.

1. **Add "args":** We added a new variable input named "args", which serves as the input variable of the general control arguments.
2. **Create a new function of "_ttest_finish()"** This function follows the one used in Scipy (Virtanen et al., 2020), which takes in the correlation matrix and degree of freedom to calculate the t-statistics and p-values.
3. **AUROC calculation:** We created a new function to calculate the AUROC results of the reconstructed structure. The implementation is done with the help of "roc_auc_score()" from sklearn (Pedregosa et al., 2011).

**Experimental Setting.** We run SIDEC on a single NVIDIA Tesla V100 SXM2 graphic card, which has 32 GB of graphic memory and 5120 NVIDIA CUDA Cores. The batch size is set as 256, but we believe it can be larger to even 2048. The learning rate is set as $0.0005$. The maximum epochs for training is 500.

**Minor Impact on Running Time.** Our method, SIDEC, integrates structural inference through two main components: training of VDE and subsequent inference utilizing both VDE and PCOR. Through our experimental observations, we found that the training phase of VDE is the most time-consuming aspect, accounting for approximately 98% of the total computation time. This pattern is consistent across all our experiments.

## B  MORE DETAILS ABOUT DATASETS

**StructInfer benchmark.** The StructInfer benchmark (Anonymous, 2023) evaluated 12 structural inference methods in a comprehensive way on a synthetic dataset. The dataset is firmly believed as created by the author, and it covers 11 types of different underlying interaction graphs and two types of dynamical simulations. We name the dataset StructInfer, which is the same as the name provided on the official website `https://structinfer.github.io/`. As there are so many trajectories, we chose the ones under the name "Vascular Networks", or in short "VN", whose underlying interaction graphs approximate the real-world vascular networks in bio-systems. Besides, we also contact the authors and get the trajectories with more than 100 nodes, and the ones with additive Gaussian noise. We would like to thank the authors for their help. As the data is already

split into three sets: for training, for validation, and for testing, we keep this setting. In the following paragraphs, we describe more details about the Springs and NetSims simulations utilized by the StructInfer benchmark.

For **Springs** simulation, it follows the approach by (Kipf et al., 2018), to simulate spring-connected particles' motion in a 2D box using the Springs simulation. In this setup, nodes represent particles, and edges correspond to springs governed by Hooke's law. The Springs simulation's dynamics are described by a second-order ordinary differential equation: $m_i \cdot x_i''(t) = \sum_{j \in \mathcal{N}_i} -k \cdot (x_i(t) - x_j(t))$. Here, $m_i$ represents particle mass (assumed as 1), $k$ is the fixed spring constant (set to 1), and $\mathcal{N}_i$ is the set of neighboring nodes with directed connections to node $i$, which is sub-sampled from the graphs generated in the StructInfer in previous steps. We integrate this equation to compute $x_i'(t)$ and subsequently $x_i(t)$ for each time step $t$. The resulting values of $x_i'(t)$ and $x_i(t)$ create 4D node features at each time step.

For **NetSims** simulation, it is firstly mentioned in NetSim dataset (Smith et al., 2011), which offers simulations of blood-oxygen-level-dependent (BOLD) imaging data in various human brain regions. Nodes in the dataset represent spatial regions of interest from brain atlases or functional tasks. Interaction graphs from the previous section determine connections between these regions. Dynamics are governed by a first-order ODE model: $x_i'(t) = \sigma \cdot \sum_{j \in \mathcal{N}_i} x_j(t) - \sigma \cdot x_i(t) + C \cdot u_i$, where $\sigma$ controls temporal smoothing and neural lag (set to 0.1 based on (Smith et al., 2011), and $C$ regulates external input interactions (set to zero to minimize external input noise) (Smith et al., 2011). 1D node features at each time step are obtained from the sampled $x_i(t)$.

For **trajectories with Gaussian noise**, we refer to the ones from StructInfer dataset (Anonymous, 2023), which we utilized in our study, includes trajectories with Gaussian noise added at various levels. As detailed in the dataset's documentation, the authors introduced this noise to the generated trajectories to simulate real-world conditions more accurately. The modified node features, represented as $\tilde{v}_i^t$, are calculated using the formula: $\tilde{v}_i^t = v_i^t + \zeta \cdot 0.02 \cdot \Delta$, where $\zeta$ is a standard Gaussian random variable ($\zeta \sim \mathcal{N}(0, 1)$), $v_i^t$ denotes the original feature vector of node $i$ at time $t$, and $\Delta$ represents the designated noise level. The noise levels applied range from 1 to 5 across all the original trajectories.

**Synthetic networks.** The six directed Boolean networks (LI, LL, CY, BF, TF, BF-CV) are the most often observed fragments in many gene regulatory networks, each has 7, 18, 6, 7, 8 and 10 nodes, respectively. Thus by carrying out experiments on these networks, we can acknowledge the performance of the chosen methods on the structural inference of real-world biological networks. We collect the six ground-truth directed Boolean networks from Pratapa et al. (2020) and simulate the single-cell evolving trajectories with BoolODE Pratapa et al. (2020) (`https://github.com/Murali-group/BoolODE`) with default settings mentioned in that paper for every network. We first sample a total number of 12000 raw trajectories. We then sample different numbers of trajectories from raw trajectories and randomly group them into three datasets: for training, for validation, and for testing, with a ratio of $8 : 2 : 2$. After that, we sample different numbers of snapshots according to the requirements of experiments in Section 5.1 with equal time intervals in every trajectory and save them as ".npy" files for data loading.

## C    IMPLEMENTATION OF BASELINES

### C.1    NRI

We use the official implementation code by the author from `https://github.com/ethanfetaya/NRI` with a customized data loader for our chosen datasets. We add our metric evaluation in the "test" function, after the calculation of accuracy in the original code.

### C.2    MPM

We use the official implementation code by the author from `https://github.com/hilbert9221/NRI-MPM` with a customized data loader for our chosen datasets. We add our metric evaluation for AUROC in the "evaluate()" function of class "XNRIDECIns" in the original code.

## C.3   ACD

We follow the official implementation code by the author as the framework for ACD (`https://github.com/loeweX/AmortizedCausalDiscovery`). We run the code with a customized data loader for the datasets in this work. We implement the metric-calculation pipeline in the "forward_pass_and_eval()" function.

## C.4   ISIDG

We ask the authors of iSIDG for the code and follow the instructions on the settings of hyper-parameters in their work. We disable the metric evaluations for the AUPRC and Jaccard index in the original implementation of iSIDG for faster computation.

## C.5   RCSI

We would like to thank the authors of RCSI for the code. We follow the instructions on the settings of hyper-parameters in their work. Same as iSIDG, we disable the metric evaluations for AUPRC and Jaccard index in the original implementation of iSIDG for faster computation.

## C.6   TIGRESS

In our utilization of TIGRESS, we adhere to the official implementation provided by the author, accessible at `https://github.com/jpvert/tigress`. To streamline our experimental workflow, we have developed a specialized wrapper. This wrapper exhibits the capability to parse multiple arguments, thereby facilitating the selection of specific trajectories for focused inference. Additionally, it seamlessly transforms trajectories into a format compatible with TIGRESS, orchestrates the systematic input of trajectories into the TIGRESS algorithm, and efficiently archives the resultant outputs in predefined directories.

## C.7   ARACNE

We use the implementation of ARACNe by the Bioconductor (Huber et al., 2015) package `minet` (Meyer et al., 2008) with a customized wrapper for the selection of a set of targeted trajectories for inference, transform trajectories into a suitable format, feed each trajectory into the ARACNe algorithm, and store the output into designated directories.

## C.8   CLR

For our implementation of CLR, we harness the capabilities of the `minet` package, a Bioconductor resource (Huber et al., 2015; Meyer et al., 2008). To optimize our workflow, we have developed a tailored wrapper. This wrapper has the ability to parse multiple arguments, enabling the selection of specific trajectories for targeted inference. Furthermore, it facilitates the seamless transformation of trajectories into a compatible format, orchestrates the sequential input of trajectories into the CLR algorithm, and efficiently stores the resulting outputs in designated directories.

## C.9   PIDC

We use the official implementation of PIDC by the author at `https://github.com/Tchanders/NetworkInference.jl` with a customized wrapper for data loading. Our wrapper will parse multiple arguments to select a set of targeted trajectories for inference, transform trajectories into a suitable format, feed each trajectory into the PIDC algorithm, and store the output into designated directories.

## C.10   SCRIBE

Our utilization of Scribe relies on the official implementation provided by the developer, which is accessible at `https://github.com/aristoteleo/Scribe-py`. To streamline our workflow, we have developed a custom wrapper. This wrapper is designed to perform a range of tasks,

including the selection of specific target trajectories for inference, the transformation of trajectories into a compatible format, the sequential input of trajectories into the Scribe algorithm, and the systematic organization and storage of output results within designated directories.

## C.11 DYNGENIE3

Our usage of dynGENIE3 relies on the official implementation authored by the developer, accessible at `https://github.com/vahuynh/dynGENIE3`. To streamline the process, we have designed a custom wrapper. This wrapper performs various tasks, such as the selection of target trajectories for inference, data format conversion to ensure compatibility, sequential input of trajectories into the XGBGRN algorithm, and systematic storage of output results in designated directories.

## C.12 XGBGRN

We employ the official XGBGRN implementation developed by the author, accessible at `https://github.com/lab319/GRNs_nonlinear_ODEs`, within a customized wrapper. This wrapper serves multiple purposes, including the selection of specific trajectories for inference, the transformation of these trajectories into a compatible format, the sequential feeding of each trajectory into the XGBGRN algorithm, and the organized storage of output results in dedicated directories.

## D    MORE EXPERIMENTAL RESULTS

Because of the page limitation, more experimental results can be found in this section.

### D.1    PERFORMANCE ON RANDOM GRAPHS

One of the key advantages of SIDEC lies in its absence of a uniform prior on edges, distinguishing it from previous VAE-based structural inference methods (Kipf et al., 2018; Chen et al., 2021; Löwe et al., 2022; Wang & Pang, 2022; Wang et al., 2023a), which incorporate a uniform prior in their KL-divergence terms. This distinction prompted our interest in assessing SIDEC and the baseline methods on trajectories where edges possess a fixed probability of existence or absence. For this purpose, we selected the "Social_Networks" trajectories from the StructInfer benchmark, characterized by underlying graphs generated via Erdős-Rényi graph (ERDdS & R&wi, 1959) generators, featuring directed edges generated with a fixed probability.

Our experimentation involved running SIDEC and the baseline methods on "SN_SP" and "SN_NS" trajectories, each with 15, 30, 50, and 100 nodes. The results, presented as the average AUROC values from ten runs, are detailed in Table 5. A noteworthy observation from the table is that when tasked with structural inference on trajectories produced by Springs simulations, featuring four-dimensional features, the performance differences between SIDEC and VAE-based methods are less pronounced compared to the results outlined in Section 5.2.

This phenomenon is particularly evident when the graph is relatively small, containing no more than 30 nodes. This aligns with the notion that VAE-based structural inference methods, equipped with the assumed uniform prior and the capacity to extract information from multi-dimensional features, exhibit strong performance in scenarios where the underlying interaction graph resembles an Erdős-Rényi graph and the trajectories exhibit multi-dimensional characteristics.

However, as the graph size increases, SIDEC consistently outperforms VAE-based methods, benefiting from its scale-insensitive nature. Furthermore, when trajectories solely comprise one-dimensional features, SIDEC surpasses VAE-based methods significantly. In such cases, the incorporation of VDE for dynamic encoding, in conjunction with the synergy between PCOR and VDE, equips SIDEC to effectively address information deficiencies within the trajectories.

Table 5: Average AUROC Results (%) of SIDEC and baselines on SN trajectories.

| Methods | SN_SP_15 | SN_SP_30 | SN_SP_50 | SN_SP_100 |
|---|---|---|---|---|
| NRI | $93.3_{\pm 0.01}$ | $79.9_{\pm 0.02}$ | $80.4_{\pm 0.02}$ | $71.8_{\pm 0.01}$ |
| MPM | $92.7_{\pm 0.00}$ | $79.3_{\pm 0.01}$ | $75.9_{\pm 0.01}$ | $69.4_{\pm 0.03}$ |
| ACD | $93.5_{\pm 0.01}$ | $81.2_{\pm 0.01}$ | $79.6_{\pm 0.02}$ | $68.8_{\pm 0.02}$ |
| iSIDG | $93.5_{\pm 0.00}$ | $81.4_{\pm 0.01}$ | $80.8_{\pm 0.02}$ | $69.3_{\pm 0.01}$ |
| RCSI | $\mathbf{94.1}_{\pm 0.02}$ | $82.7_{\pm 0.01}$ | $81.3_{\pm 0.01}$ | $73.4_{\pm 0.02}$ |
| SIDEC | $94.0_{\pm 0.02}$ | $\mathbf{89.8}_{\pm 0.01}$ | $\mathbf{89.0}_{\pm 0.04}$ | $\mathbf{90.0}_{\pm 0.01}$ |

| Methods | SN_NS_15 | SN_NS_30 | SN_NS_50 | SN_NS_100 |
|---|---|---|---|---|
| NRI | $58.4_{\pm 0.04}$ | $51.4_{\pm 0.01}$ | $49.6_{\pm 0.03}$ | $50.2_{\pm 0.03}$ |
| MPM | $67.4_{\pm 0.02}$ | $50.9_{\pm 0.01}$ | $53.1_{\pm 0.03}$ | $50.1_{\pm 0.02}$ |
| ACD | $65.2_{\pm 0.05}$ | $52.9_{\pm 0.03}$ | $49.3_{\pm 0.02}$ | $50.8_{\pm 0.01}$ |
| iSIDG | $66.1_{\pm 0.04}$ | $53.8_{\pm 0.03}$ | $54.8_{\pm 0.01}$ | $51.7_{\pm 0.02}$ |
| RCSI | $67.6_{\pm 0.03}$ | $55.8_{\pm 0.02}$ | $55.0_{\pm 0.02}$ | $53.2_{\pm 0.03}$ |
| SIDEC | $\mathbf{91.6}_{\pm 0.02}$ | $\mathbf{90.2}_{\pm 0.02}$ | $\mathbf{89.7}_{\pm 0.02}$ | $\mathbf{89.2}_{\pm 0.02}$ |

### D.2 PERFORMANCE ON UNDIRECTED GRAPHS

As established at the outset of this paper, numerous correlation-based structural inference methods are ill-suited for the reconstruction of directed graphs. While our prior experiments exclusively employed trajectories with directed underlying interaction graphs, it's equally pertinent to assess the performance of correlation-based techniques, traditional baselines, and our proposed SIDEC on trajectories generated with undirected underlying interaction graphs. Fortunately, the StructInfer benchmark contains such trajectories, specifically under the category "Landscape_Networks," featuring undirected underlying interaction graphs. In addition to the aforementioned baselines, we have incorporated several other methods for comparative evaluation:

- TIGRESS (Haury et al., 2012): A correlation-based model employing iterative feature selection with least-angle regression and bootstrapping.
- ARACNe (Margolin et al., 2006): A mutual-information-based method that calculates pairwise mutual information and utilizes the Data Processing Inequality principle to eliminate indirect interactions.
- CLR (Faith et al., 2007): This method employs pairwise mutual information but assumes a background noise distribution for mutual information.
- PIDC (Chan et al., 2017): A model based on Partial Information Decomposition, interpreting aggregated unique information as the strength of edges between nodes.
- Scribe (Qiu et al., 2020): Utilizes Restricted Directed Information (Rahimzamani & Kannan, 2016) and its variants to quantify causality within the structure by considering the influence of confounding factors.
- dynGENIE3 (Huynh-Thu & Geurts, 2018): A random-forest-based method that employs ordinary differential equations to model time series dynamics.
- XGBGRN (Ma et al., 2020): Similar to dynGENIE3 but leverages XGBoost (Chen & Guestrin, 2016) instead of random forests.

It's essential to note that these methods operate exclusively on one-dimensional features and cannot perform structural inference on trajectories generated using Springs simulation. Additionally, their design inherently prevents them from inferring directed edges. In our evaluation, we've also implemented an algorithm based solely on partial correlation coefficients, denoted as PCOR, and a variant of SIDEC that lacks the component for rearranging latent variables to reconstruct directed graphs, known as SIDEC-V. Besides that, we also implemented following correlation-based structural inference methods: Person Correlation (P-cor) (Maucher et al., 2011), Distance Correlation (D-cor) (Liu et al., 2021), Low-order Partial Correlation (LPCOR) (Zuo et al., 2014), and Semi-partial Correlation Coefficients (SPCOR) (Kim, 2015). These methods, by design, only produce symmetric results. We show the results in Table. 6. On "LN_SP" trajectories, SIDEC and SIDEC-V consistently outperform VAE-based methods, displaying a robustness independent of graph size, with minor drops in AUROC values. This trend aligns with our findings in Section 5.2.

However, when examining the results for "LN_NS" trajectories, SIDEC exhibits superior performance mainly on graphs with no fewer than 30 nodes. It's worth noting that the performance

Table 6: Average AUROC Results (%) of SIDEC and baselines on LN trajectories.

| Methods | LN_SP_15 | LN_SP_30 | LN_SP_50 | LN_SP_100 |
|---------|----------|----------|----------|-----------|
| NRI | $97.0\pm 0.02$ | $94.9\pm 0.00$ | $87.1\pm 0.01$ | $82.8\pm 0.01$ |
| MPM | $97.9\pm 0.01$ | $95.5\pm 0.02$ | $86.9\pm 0.01$ | $84.2\pm 0.03$ |
| ACD | $97.0\pm 0.02$ | $95.8\pm 0.01$ | $87.6\pm 0.02$ | $83.9\pm 0.02$ |
| iSIDG | $97.4\pm 0.02$ | $94.7\pm 0.02$ | $87.4\pm 0.02$ | $83.2\pm 0.02$ |
| RCSI | $97.3\pm 0.02$ | $94.4\pm 0.02$ | $88.0\pm 0.02$ | $84.3\pm 0.02$ |
| SIDEC-V | $97.6\pm 0.02$ | $96.3\pm 0.02$ | $96.0\pm 0.02$ | $94.7\pm 0.02$ |
| SIDEC | $\mathbf{97.7}\pm \mathbf{0.02}$ | $\mathbf{96.5}\pm \mathbf{0.03}$ | $\mathbf{96.2}\pm \mathbf{0.02}$ | $\mathbf{95.1}\pm \mathbf{0.03}$ |

| Methods | LN_NS_15 | LN_NS_30 | LN_NS_50 | LN_NS_100 |
|---------|----------|----------|----------|-----------|
| TIGRESS | $84.1\pm 1.16$ | $87.4\pm 3.32$ | $92.2\pm 0.42$ | $93.8\pm 1.96$ |
| ARACNe | $92.3\pm 4.84$ | $80.4\pm 5.67$ | $71.2\pm 0.48$ | $62.8\pm 8.36$ |
| CLR | $97.3\pm 3.17$ | $96.6\pm 4.87$ | $91.0\pm 2.35$ | $95.0\pm 0.53$ |
| PIDC | $\mathbf{97.5}\pm \mathbf{1.01}$ | $82.0\pm 7.28$ | $88.6\pm 1.69$ | $94.2\pm 2.28$ |
| Scribe | $54.2\pm 3.98$ | $56.2\pm 3.88$ | $52.1\pm 2.49$ | $52.5\pm 1.62$ |
| dynGENIE | $51.3\pm 5.21$ | $50.1\pm 2.42$ | $50.5\pm 1.22$ | $67.3\pm 14.23$ |
| XGBGRN | $97.2\pm 1.13$ | $96.5\pm 2.10$ | $96.9\pm 0.83$ | $\mathbf{98.0}\pm \mathbf{0.93}$ |
| NRI | $56.0\pm 0.04$ | $53.9\pm 0.02$ | $54.4\pm 0.02$ | $51.8\pm 0.03$ |
| MPM | $52.2\pm 0.02$ | $62.1\pm 0.05$ | $53.4\pm 0.01$ | $50.4\pm 0.03$ |
| ACD | $61.9\pm 0.03$ | $61.6\pm 0.04$ | $53.4\pm 0.02$ | $50.2\pm 0.02$ |
| iSIDG | $59.2\pm 0.05$ | $56.2\pm 0.03$ | $55.7\pm 0.03$ | $52.3\pm 0.02$ |
| RCSI | $60.3\pm 0.03$ | $60.2\pm 0.02$ | $57.6\pm 0.03$ | $53.5\pm 0.02$ |
| P-cor | $68.9\pm 0.31$ | $65.1\pm 0.10$ | $66.7\pm 0.39$ | $68.1\pm 0.44$ |
| D-cor | $67.3\pm 0.23$ | $66.4\pm 0.37$ | $65.2\pm 0.20$ | $65.0\pm 0.26$ |
| LPCOR | $81.9\pm 0.60$ | $80.6\pm 1.56$ | $78.5\pm 0.98$ | $69.5\pm 3.01$ |
| SPCOR | $95.0\pm 0.19$ | $91.8\pm 3.08$ | $84.0\pm 0.63$ | $71.5\pm 2.55$ |
| PCOR | $97.3\pm 0.56$ | $92.0\pm 5.20$ | $84.8\pm 1.66$ | $79.2\pm 4.32$ |
| SIDEC-V | $97.2\pm 0.03$ | $93.8\pm 0.03$ | $96.0\pm 0.01$ | $96.0\pm 0.02$ |
| SIDEC | $97.4\pm 0.03$ | $\mathbf{96.7}\pm \mathbf{0.02}$ | $\mathbf{97.0}\pm \mathbf{0.03}$ | $97.5\pm 0.02$ |

gaps are relatively narrow, with XGBGRN, CLR, and PIDC maintaining high accuracy across the datasets. Nonetheless, these methods display substantial result deviations, indicating limited stability and reliability in reconstruction. Furthermore, XGBGRN, CLR, and PIDC cannot accommodate trajectories with multi-dimensional features, as they conflict with their core operating mechanisms. The results indicate that both P-cor and D-cor show subpar performance across all datasets. This can be attributed to their limitation of considering only pairwise correlations and overlooking the influence of other potentially impactful nodes. LPCOR, while better, still falls short as it calculates coefficients conditional on a subset rather than all other variables.

Interestingly, SPCOR and PCOR demonstrate comparable performance, with PCOR slightly outperforming SPCOR. This can be attributed to SPCOR's reliance on semi-partial correlation calculations, which may limit its effectiveness in identifying connections between nodes with larger degrees. Despite the variations in graph structure across the datasets, neither SPCOR nor PCOR outperforms SIDEC. In contrast, SIDEC boasts broader applicability, irrespective of node feature dimensionality, and consistently delivers medium to high accuracy. In comparison to PCOR, which can be regarded as the second half of SIDEC, our proposed method markedly enhances structural reconstruction accuracy, underscoring the value of encoding node dynamics in structural inference.

### D.3 Performance on Larger Graphs

To bolster our claim of SIDEC's scalability, we conducted additional experiments using the NRI simulator (Kipf et al., 2018) – a tool also employed by (Chen et al., 2021; Löwe et al., 2022; Wang & Pang, 2022; Wang et al., 2023a; Wang & Pang, 2023) for dataset generation. These experiments involved generating spring-ball systems with 500 and 1,000 balls, producing a total of 12,000 trajectories per graph, subsequently divided into training, validation, and test sets in an 8:2:2 ratio.

In these experiments, only SIDEC was able to operate effectively without encountering out-of-memory (OOM) issues, unlike other VAE-based baseline methods. The resulting average AUROC values over ten runs are shown in Table 7. Compared with other methods, SIDEC managed to re-

Table 7: Average AUROC Results (%) of SIDEC on spring-ball systems.

|  | w. 500 Balls | w. 1,000 Balls |
|---|---|---|
| SIDEC | $92.8_{\pm 0.04}$ | $90.1_{\pm 0.03}$ |

Table 8: Statistics of PEMS datasets.

| Dataset | # Nodes | # Edges | # Time Steps | Missing Ratio |
|---|---|---|---|---|
| PEMS03 | 358 | 547 | $26,208$ | 0.672% |
| PEMS04 | 307 | 340 | $16,992$ | 3.182% |
| PEMS07 | 883 | 866 | $28,224$ | 0.452% |

Table 9: Average AUROC Results (%) of SIDEC on PEMS datasets.

|  | PEMS03 | PEMS04 | PEMS07 |
|---|---|---|---|
| SIDEC | $72.8_{\pm 0.07}$ | $75.4_{\pm 0.04}$ | $70.1_{\pm 0.06}$ |

construct the structure of these graphs with 500 and 1,000 nodes. And compared with the results of it on smaller graphs, the results on the AUROC values still support its high accuracy on the structural inference. These results not only demonstrate SIDEC's capability to reconstruct the structure of large graphs but also maintain high accuracy in structural inference across varying graph sizes.

### D.4 Performance on Real-world Dataset

We would like to emphasis that obtaining comprehensive, real-world data for structural inference poses significant challenges, both in terms of cost and time. Often, real-world data lack complete information on node features or reliable knowledge of underlying interaction graphs, as seen in gene regulatory networks (Pratapa et al., 2020) and scene graphs (Yang et al., 2018). In other cases, such as transaction networks (Tao et al., 2021) and road maps (Biagioni & Eriksson, 2012), data may not cover all nodes in the system.

Despite these challenges, we agree with the necessity of testing our method, SIDEC, on real-world data. To this end, we have conducted additional experiments using three widely recognized public traffic network datasets: PEMS03, PEMS04, and PEMS07 (Song et al., 2020). These datasets, derived from the California Caltrans Performance Measurement System (PeMS) (Chen et al., 2001), comprise data aggregated into 5-minute intervals. The adjacency matrix of the nodes is constructed by road network distance with a thresholded Gaussian kernel (Song et al., 2020). Table 8 summarizes these datasets.

We resampled the data such that constructing 49 time steps of points for each trajectory, and obtained 534, 346, and 547 trajectories, respectively. We then split all datasets with ratio 8:2:2 into training sets, validation sets and test sets, and run SIDEC on all of them. Our results, presented in Table 9, demonstrate SIDEC's effective performance despite challenges such as noise, missing data, and adjacency matrix inaccuracies. It's important to note that these datasets' adjacency matrices only connect sensors on the same road, omitting alternative connecting paths, which could impact results. As shown in the table, despite the noise and missing data, and the inaccuracy in the adjacency matrices provided by the dataset, SIDEC manages to perform well and can deal with the challenges. It is notable that the adjacency matrices in these datasets just connect the sensors if they are on the same road, but they ignore the existence of paths or alleys that may also connect the sensors. Because of this, when compared with the experimental results discussed in Section 5, the performance of SIDEC is impacted.

## E Limitations

While SIDEC showcases considerable capabilities in structural inference, akin to other methods in this domain, it does have certain limitations that warrant attention.

First and foremost, SIDEC is tailored for dynamical systems characterized by a static underlying interacting structure. In scenarios where the underlying structure undergoes temporal variations,

the simplicity of the VDE setup can hinder the effective capture of node dynamics. However, we envision the possibility of extending SIDEC to accommodate dynamic graphs by encoding edge dynamics, thus addressing this limitation.

Secondly, the dynamical systems under investigation must be fully observed within the defined time period. This entails that all nodes need to be observed, and their features meticulously recorded. In principle, the VDE operates on a Markovian assumption, which could be theoretically compromised when some influential factors remain unobserved. Nevertheless, we maintain confidence in SIDEC's adaptability to such tasks, as it primarily relies on node features, potentially mitigating these concerns.

Thirdly, our evaluation of SIDEC was exclusively conducted on synthetic datasets, leaving a void in the examination of its performance on real-world, intricate data. This highlights the pressing need for a comprehensive and reliable real-world structural inference dataset, an omission in the existing landscape. However, it's important to acknowledge that these limitations serve as catalysts for future research endeavors aimed at enhancing both SIDEC and structural inference methodologies in general. By surmounting these challenges, we can substantially expand the application domains of SIDEC, offering invaluable contributions to researchers across diverse fields.

## F  BROADER IMPACT

Much like NRI, MPM, ACD, iSIDG, and other structural inference methodologies, SIDEC extends its utility to a diverse range of researchers across the realms of physics, chemistry, sociology, and biology. In our investigations, we have demonstrated SIDEC's proficiency in reconstructing expansive graph structures while displaying robustness to variations in the dimensionality of node features, underscoring its versatility and broad applicability. However, the advent of structural inference technology, while profoundly beneficial to many fields, also raises concerns about potential misuse. There exists the possibility of utilizing this technology to unveil intimate relationships among users within social networks, which, if misapplied, could encroach upon individuals' privacy.

