# OpenReview forum: "Structural Inference with Dynamics Encoding and Partial Correlation Coefficients"
_ICLR.cc/2024/Conference — ICLR 2024 poster_

### Official Review · Reviewer_LxHJ · 2023-10-13

**Soundness:** 4 excellent
**Presentation:** 4 excellent
**Contribution:** 4 excellent
**Rating:** 10
**Confidence:** 4

**Summary:**

The paper introduces an innovative approach to structural inference (SIDEC), employing dynamics encoding and partial correlation coefficients. The authors employ variational inference to encode node dynamics within latent variables and reorganize these variables temporally to reconstruct directed graph structures. The resulting method stands out for its scalability, accuracy, and versatility, outperforming existing approaches as demonstrated through experimental results. The paper additionally includes a comprehensive implementation appendix and provides a link to the anonymous GitHub repository hosting the SIDEC codebase, ensuring reproducibility.

**Strengths:**

1. The paper presents a novel approach to structural inference, combining variational dynamics encoding with partial correlation coefficients, demonstrating superior performance compared to existing methods, as validated by experimental results. The use of a variational model to encode node dynamics addresses scalability challenges from previous methods, and this application of Variational Autoencoders (VAE) in structural inference is a notable contribution.

2. The method exhibits impressive scalability, accommodating trajectories with both one-dimensional and multi-dimensional features, making it applicable to a wide range of scenarios.

3. The paper is exceptionally well-organized and articulated, employing clear and concise language throughout.

4. The authors provide a detailed explanation of their proposed method, including mathematical formulations and implementation details, enhancing its accessibility and reproducibility.

5. The evaluation is meticulously crafted and comprehensive, successfully establishing the superior performance of SIDEC in comparison to existing methods. The additional experimental results presented in the appendix further bolster the credibility and persuasiveness of SIDEC's superiority.

6. The limitations of the proposed method are conscientiously addressed in the appendix.

**Weaknesses:**

While the paper is commendable overall, one potential weakness lies in the absence of an evaluation on real-world data. The authors acknowledge this in the "Limitations" section, citing the challenges of collecting such data. It would be beneficial to include plans for addressing this in future work.

**Questions:**

1. The choice of using partial correlation as the final step in reconstructing the graph structure merits further clarification.

2. A suggestion for refining the paper layout is to consider rearranging Algorithms 1 and 2 before presenting Algorithms 3 and 4 in the appendix for better logical flow.

---

> ### Author Response · Authors · 2023-11-15
> **Response to Reviewer LxHJ**
>
> We are grateful to the reviewer for their encouraging and insightful feedback. It is heartening to hear that the novelty of the SIDEC concept and its broad applicability were well-received, alongside the paper's organized presentation of both theory and implementation. The acknowledgment of SIDEC's superior performance compared to baseline models is also greatly appreciated.
>
> Regarding the concerns raised by the reviewer:
>
> > **W.** While the paper is commendable overall, one potential weakness lies in the absence of an evaluation on real-world data. The authors acknowledge this in the "Limitations" section, citing the challenges of collecting such data. It would be beneficial to include plans for addressing this in future work.
>
> Thank you for highlighting this issue. While we did add experiments on three real-world datasets in response to Reviewer 5DvY, these datasets proved problematic due to their unreliable and imprecise adjacency matrices, which compromised our ability to fully showcase SIDEC's strengths and performance.
>
> We used the PEMS03, PEMS04, and PEMS07 datasets [1] from California's Caltrans Performance Measurement System (PeMS) [2]. All these data are aggregated into 5-minutes windows, which means there are 288 points in the traffic flow for one day. The adjacency matrix of the nodes is constructed by road network distance with a thresholded Gaussian kernel [1]. The summary of these datasets can be found in the following table:
>
> | Datasets | # Nodes | # Edges | # TimeSteps | # MissingRatio |
> | -------- | ------- | ------- | ----------- | -------------- |
> | PEMS03   | 358     | 547     | 26208       | 0.672\%        |
> | PEMS04   | 307     | 340     | 16992       | 3.182\%        |
> | PEMS07   | 883     | 866     | 28224       | 0.452\%        |
>
> We resampled the data such that constructing 49 time steps of points for each trajectory, and obtained 534, 346, and 547 trajectories, respectively. We then split all datasets with ratio 8:2:2 into training sets, validation sets and test sets, and run SIDEC on all of them. We report the average AUROC values (in \%) of ten runs in the following table:
>
> |       | PEMS03            | PEMS04            | PEMS07            |
> | ----- | ----------------- | ----------------- | ----------------- |
> | SIDEC | $ 72.8 \pm 0.07 $ | $ 75.4 \pm 0.04 $ | $ 70.1 \pm 0.06 $ |
>
> As shown in the table, despite the noise and missing data, and the inaccuracy in the adjacency matrices provided by the dataset, SIDEC manages to perform well and can deal with the challenges. It is notable that the adjacency matrices in these dataset just connect the sensors if thy are on the same road, but they ignore the existence of paths or alleys that may also connect the sensors. Because of this, when compared with the experimental results discussed in the original submission, the performance of SIDEC decreases massively.
>
> We've added these findings to the Appendix D.4 of our revised manuscript. We believe this additional real-world data testing not only addresses your concerns but also significantly strengthens the validity and applicability of our research. We plan to address this by collecting more reliable gene regulatory network data through in-lab experiments, although this is a time-intensive process.
>
> ------------------
>
> > **Q1.** The choice of using partial correlation as the final step in reconstructing the graph structure merits further clarification.
>
> Many thanks for the comment. We selected the partial correlation coefficient (PCOR) as it has shown superior performance in StructInfer [3] benchmarks, especially for one-dimensional trajectories. We believe that with effective dimension extraction and dynamic embedding, PCOR can be adapted for broader applications, including directed graphs and multi-dimensional nodes. This was a key factor in developing SIDEC.
>
> ------------
>
> > **Q2.** A suggestion for refining the paper layout is to consider rearranging Algorithms 1 and 2 before presenting Algorithms 3 and 4 in the appendix for better logical flow.
>
> Thank you for this suggestion. We have revised the manuscript accordingly to enhance readability and logical flow.
>
> We hope our responses satisfactorily address the reviewer's concerns, and we extend our gratitude for their valuable contributions toward improving our paper. To facilitate easy identification, sections and paragraphs that have been revised in the manuscript are highlighted in blue.
>
> -----
> ### References
>
> [1] C. Song, Y. Lin, S. Guo and H. Wan.Spatial- Temporal Synchronous Graph Convolutional Networks: A New Framework for Spatial-Temporal Network Data Forecasting. AAAI, volume 34, 914–921.
>
> [2] C. Chen, K. Petty, A. Skabardonis, P. Varaiya and Z. Jia. Freeway performance measurement system: mining loop detector data. Transportation Research Record 1748(1):96–102.
>
> [3] Authors Anonymous. Benchmarking structural inference methods for dynamical interacting systems. https://structinfer.github.io/, 2023.

---

> > ### Comment · Reviewer_LxHJ · 2023-11-21
> > **Thank you for response!**
> >
> > Dear Authors,
> >
> > Thank you for your detailed and enlightening response to the review, as well as the responses to other reviewers. The additional information and clarifications provided regarding the SIDEC concept, particularly its application to real-world datasets and the choice of using partial correlation, have significantly enhanced the paper's depth and clarity.
> >
> > I appreciate the effort to include experiments on the PEMS datasets, despite the challenges with their adjacency matrices. Your thorough analysis and the consequent addition to Appendix D.4 effectively address the concern regarding real-world data evaluation. This not only resolves the issue raised but also substantially strengthens the paper's practical applicability and validity.
> >
> > Furthermore, your clarification on the selection of partial correlation coefficient and the rearrangement of algorithms in the manuscript for improved logical flow are commendable. These adjustments align well with the paper’s objectives and contribute to its overall coherence and readability.
> >
> > Considering these comprehensive enhancements and the proactive steps taken to address the previously mentioned concerns, I am convinced of the paper's increased value and significance. Therefore, I am inclined to revise my evaluation score upwards, reflecting the improvements and the robustness added to the paper through your response.
> >
> > Warm regards,
> >
> > Reviewer LxHJ

---

> > > ### Author Response · Authors · 2023-11-21
> > >
> > > Dear Reviewer,
> > >
> > > I sincerely appreciate your kind words and the upward revision of your evaluation score. Your recognition of the efforts we put into addressing the concerns and enhancing the paper is very encouraging. Thank you once again for your thoughtful and constructive review. It has not only helped improve our paper but has also provided us with valuable insights for our future work.
> > >
> > > Warm regards,
> > > Authors

---

### Official Review · Reviewer_5DvY · 2023-10-30

**Soundness:** 2 fair
**Presentation:** 2 fair
**Contribution:** 2 fair
**Rating:** 6
**Confidence:** 3

**Summary:**

This paper proposes a new model for structure inference in dynamical systems, called SIDEC.
It combines two methods, the Variational Dynamics Encoder (VDE) and the Partial Correlation (PCOR), to infer the directed graph structure of a dynamical system from its trajectory data. The method captures the minimum information of node feature dynamics and can effectively reduce feature dimensions. It is applicable to time-series data with multi-dimensional and one-dimensional features. It evaluates SIDEC on 20 datasets and compares it with other benchmark methods. The results show that SIDEC outperforms other methods in terms of average AUROC metric and it is robust to Gaussian noise.

**Strengths:**

The paper introduces a novel model, SIDEC, that combines two methods, VDE and PCOR, to infer the directed graph structure of a dynamical system from its trajectory data.
The paper provides a comprehensive introduction, background, and motivation for the problem, which make it easier to follow.
It seems that SIDEC can capture the minimal information of node feature dynamics and effectively reduce feature dimensions, leverage temporal information to infer directed edges using conditional correlations, and perform robustly against Gaussian noise .
It evaluates SIDEC on 20 datasets and compares it with other benchmark methods and the results show that SIDEC outperforms other methods in terms of average AUROC metric.

**Weaknesses:**

1.	This paper mainly designs and analyzes the SIDEC framework based on information bottleneck theory and partial correlation coefficient, but it does not give clear mathematical proofs and theoretical guarantees to explain how they influence each other.
2.	Experiment Datasets are all artificially generated and may not fully reflect the complexity and diversity of the real world. Therefore, the experimental results of this article may have certain biases and limitations, and further testing and evaluation on more real data sets are required.
3.	This method combines VDB and PCOR, but the paper does not give the impact on training time after the combination.

**Questions:**

1.	SIDEC includes PCOR, and some PROR models are also mentioned in the Structural inference with correlations and partial correlations section. Why are there no comparative experiments of these models included in the experimental section?
2.	Judging from the results of the ablation experiment, the optimal size of the time window is 1. It seems that the existence of the time window does not fully improve the performance of the model. Does this mean that it is not necessary to use VDB?
3.	Suggestion: maybe you can adjust the combination of loss to the weighted sum of the reconstruction loss and the autocorrelation loss to adjust the impact of time window，and  monitor the change of the performance.

---

> ### Author Response · Authors · 2023-11-15
> **Response to Reviewer 5DvY (Part1)**
>
> We sincerely thank you, Reviewer 5DvY, for your motivating and positive feedback on our work. We appreciate your recognition of the novelty and clarity of our paper and are encouraged by your acknowledgment of SIDEC's performance. Below, we address your concerns in detail:
>
> > **W1.** This paper mainly designs and analyzes the SIDEC framework based on information bottleneck theory and partial correlation coefficient, but it does not give clear mathematical proofs and theoretical guarantees to explain how they influence each other.
>
> Thank you for your insightful feedback. We acknowledge the reviewer's concern regarding the lack of detailed mathematical proofs and theoretical guarantees in our paper, specifically in the context of the SIDEC framework's reliance on information bottleneck theory and partial correlation coefficient (PCOR).
>
> Our proposed framework, SIDEC, utilizes the variational dynamics encoder (VDE) [1], a variant of the variational autoencoder (VAE), which adheres to the variational information bottleneck (VIB) [2] principle. Specifically, the VDE in our framework seeks to optimize the encoded representation $\mathbf{z}^t_i$ to be informative about the future state $v^{t+\tau}_i$ while minimizing dependency on the current state $v^t_i$, as dictated by the information bottleneck theory [3]. This process effectively compresses the representation of input trajectories, ensuring that the dynamics of each node are captured over a defined time window $\tau$.
>
> Upon obtaining the dynamics embeddings, we utilize PCOR to infer the graph structure. PCOR's efficacy in structural inference for one-dimensional data [15], underpins its integration into our framework. The interplay between VDE and PCOR within SIDEC is thus crucial; the former captures dynamic node embeddings, while the latter leverages these embeddings for structural inference.
>
> We recognize that a comprehensive mathematical proof elaborating on this interplay is absent from our current manuscript. The decision to defer a detailed proof to future work stems from the complexity of the theoretical underpinnings, which require a more extensive treatment than our current scope allows. Future work will focus on providing these rigorous mathematical validations, particularly elucidating the mutual influence of VDE and PCOR within the SIDEC framework.
>
> We appreciate this opportunity to clarify our approach and its theoretical foundations. We believe that these additional details, along with our commitment to future work, reinforce the contributions of our paper and address the concerns raised. We will integrate this refined explanation into the Appendix of our manuscript to ensure our approach's theoretical aspects are both transparent and robust.
>
> --------------------

---

> ### Author Response · Authors · 2023-11-15
> **Response to Reviewer 5DvY (Part2)**
>
> > **W2.** Experiment Datasets are all artificially generated and may not fully reflect the complexity and diversity of the real world. Therefore, the experimental results of this article may have certain biases and limitations, and further testing and evaluation on more real data sets are required.
>
> We would like to emphasis that obtaining comprehensive, real-world data for structural inference poses significant challenges, both in terms of cost and time, which is mentioned in the section of limitation in the submission. Often, real-world data lack complete information on node features or reliable knowledge of underlying interaction graphs, as seen in gene regulatory networks [15] and scene graphs [4]. In other cases, such as transaction networks [5] and road maps [6], data may not cover all nodes in the system.
>
> Despite these challenges, we agree with the necessity of testing our method, SIDEC, on real-world data. To this end, we have conducted additional experiments using three widely recognized public traffic network datasets: PEMS03, PEMS04, and PEMS07 [7]. These datasets, derived from the California Caltrans Performance Measurement System (PeMS) [8], comprise data aggregated into 5-minute intervals. The adjacency matrix of the nodes is constructed by road network distance with a thresholded Gaussian kernel [7]. The following table summarizes these datasets:
>
> | Datasets | # Nodes | # Edges | # TimeSteps | MissingRatio |
> | -------- | ------- | ------- | ----------- | ------------ |
> | PEMS03   | 358     | 547     | 26208       | 0.672\%      |
> | PEMS04   | 307     | 340     | 16992       | 3.182\%      |
> | PEMS07   | 883     | 866     | 28224       | 0.452\%      |
>
> We resampled the data such that constructing 49 time steps of points for each trajectory, and obtained 534, 346, and 547 trajectories, respectively. We then split all datasets with ratio 8:2:2 into training sets, validation sets and test sets, and run SIDEC on all of them. Our results, presented below, demonstrate SIDEC's effective performance despite challenges such as noise, missing data, and adjacency matrix inaccuracies. It's important to note that these datasets' adjacency matrices only connect sensors on the same road, omitting alternative connecting paths, which could impact results.
>
> |       | PEMS03            | PEMS04            | PEMS07            |
> | ----- | ----------------- | ----------------- | ----------------- |
> | SIDEC | $ 72.8 \pm 0.07 $ | $ 75.4 \pm 0.04 $ | $ 70.1 \pm 0.06 $ |
>
> As shown in the table, despite the noise and missing data, and the inaccuracy in the adjacency matrices provided by the dataset, SIDEC manages to perform well and can deal with the challenges. It is notable that the adjacency matrices in these datasets just connect the sensors if they are on the same road, but they ignore the existence of paths or alleys that may also connect the sensors. Because of this, when compared with the experimental results discussed in the original submission, the performance of SIDEC is impacted. We've added these findings to Appendix D.4 of our revised manuscript. We believe this additional real-world data testing not only addresses your concerns but also significantly strengthens the validity and applicability of our research.
>
> -----------------
>
> > **W3.** This method combines VDB and PCOR, but the paper does not give the impact on training time after the combination.
>
> We appreciate your insightful observation regarding the necessity of addressing the training time in our methodology. In response to your query, it's important to clarify that PCOR does not require training and, therefore, does not directly influence the training duration of the variational dynamics encoder (VDE). Our method, SIDEC, integrates structural inference through two main components: training of VDE and subsequent inference utilizing both VDE and PCOR. Through our experimental observations, we found that the training phase of VDE is the most time-consuming aspect, accounting for approximately 98% of the total computation time. This pattern is consistent across all our experiments. We revised the manuscript to add this description in Appendix A.7.
>
> -------------------------

---

> ### Author Response · Authors · 2023-11-15
> **Response to Reviewer 5DvY (Part3)**
>
> > **Q1.** SIDEC includes PCOR, and some PROR models are also mentioned in the Structural inference with correlations and partial correlations section. Why are there no comparative experiments of these models included in the experimental section?
>
> Thank you for this question. The decision to exclude a direct comparative analysis of PROR models in the main experimental section was made due to the inherent limitations of these models. Specifically, correlation-based structural inference methods are typically constrained to graph reconstruction of undirected graphs with one-dimensional node features, significantly limiting their application scope.
>
> To address this, we conducted a thorough evaluation of various correlation-based methods, as an addition to the experimental results demonstrated in Appendix D.2. We evaluate the baselines and SIDEC on LN\_NS datasets, which is characterized by one-dimensional node feature and undirected underlying interaction graph. We add these methods for evaluation:
>
> - Person Correlation (P-cor) [9]
> - Distance Correlation (D-cor) [10]
> - Low-order Partial Correlation (LPCOR) [11]
> - Semi-partial Correlation Coefficients (SPCOR) [12]
>
> Additionally, we incorporated PCOR and SIDEC results into this analysis for a comprehensive comparison. Our results are presented in the following table:
>
> | Methods | LN\_NS\_15   | LN\_NS\_30    | LN\_NS\_50    | LN\_NS\_100  |
> | ------- | ------------ | ------------- | ------------- | ------------ |
> | P-cor   | $68.9± 0.31$ | $65.1± 0.10$  | $66.7± 0.39$  | $68.1 ±0.44$ |
> | D-cor   | $67.3± 0.23$ | $66.4± 0.37$  | $65.2± 0.20$  | $65.0± 0.26$ |
> | LPCOR   | $81.9± 0.60$ | $80.6 ± 1.56$ | $78.5 ± 0.98$ | $69.5± 3.01$ |
> | SPCOR   | $95.0± 0.19$ | $91.8± 3.08$  | $84.0± 0.63$  | $71.5± 2.55$ |
> | PCOR    | $97.3± 0.56$ | $92.0± 5.20$  | $84.8± 1.66$  | $79.2± 4.32$ |
> | SIDEC   | $97.4± 0.03$ | $96.7± 0.02$  | $97.0± 0.03$  | $97.5± 0.02$ |
>
> The results indicate that both P-cor and D-cor show subpar performance across all datasets. This can be attributed to their limitation of considering only pairwise correlations and overlooking the influence of other potentially impactful nodes. LPCOR, while better, still falls short as it calculates coefficients conditional on a subset rather than all other variables.
>
> Interestingly, SPCOR and PCOR demonstrate comparable performance, with PCOR slightly outperforming SPCOR. This can be attributed to SPCOR's reliance on semi-partial correlation calculations, which may limit its effectiveness in identifying connections between nodes with larger degrees. Despite the variations in graph structure across the datasets, neither SPCOR nor PCOR outperforms SIDEC.
>
> We believe these findings reinforce our approach of integrating VDE within SIDEC, which significantly enhances the overall performance of structural inference. We have updated our manuscript to include these comprehensive results in Appendix D.2, offering a broader context and validation for our methodological choices.
>
> -------
> > **Q2.** Judging from the results of the ablation experiment, the optimal size of the time window is 1. It seems that the existence of the time window does not fully improve the performance of the model. Does this mean that it is not necessary to use VDB?
>
> Thank you for highlighting the significance of the optimal time window size in our ablation experiments. The choice of the time window $\tau$, is indeed a critical aspect of our model, as it determines the scale at which the dynamics of the system are analyzed. The Markovian assumption inherent in our model implies that knowing the state of the system at time $\tau$ should suffice to predict its future state at time $t+\tau$, without the need to consider its earlier history [13].
>
> In the case of the StructInfer [14] datasets, which were generated with smaller time intervals and subsequently subsampled, a $\tau$ value of 1 suggests that the system behaves Markovian over these intervals. This means that past states do not significantly influence future states beyond this time window.
>
> However, it's important to note that there are scenarios where:
>
> 1. The system might not strictly adhere to Markovian dynamics, and our observations are closely spaced such that immediate past states influence future states.
> 2. The data are directly sampled from simulations at minimal time intervals.
>
> In these scenarios, a $\tau$ value larger than 1 might be more appropriate. Determining the optimal value of $\tau$ in such cases is a complex task and requires further investigation, which we plan to undertake in future research.
>
> -------

---

> > ### Author Response · Authors · 2023-11-15
> > **Response to Reviewer 5DvY (Part4)**
> >
> > > **Q3.** Suggestion: maybe you can adjust the combination of loss to the weighted sum of the reconstruction loss and the autocorrelation loss to adjust the impact of time window, and monitor the change of the performance.
> >
> > We greatly appreciate your suggestion regarding the adjustment of our loss function to a weighted sum of reconstruction and autocorrelation losses to better account for the impact of the time window.
> >
> > Indeed, we have experimented with modifying the loss function in this manner. However, we encountered challenges in achieving convergence with this approach, particularly for longer time windows. This unexpected behavior has piqued our interest, and we are actively investigating potential solutions to this problem. We aim to better understand and possibly exploit this phenomenon in the context of our model's performance over extended time windows.
> >
> > We hope these explanations satisfactorily address the concerns and suggestions of Reviewer 5DvY, and we're grateful for the insightful feedback that will guide our ongoing research. To facilitate easy identification, sections and paragraphs that have been revised in the manuscript are highlighted in blue.
> >
> > ----------------
> >
> > ### References
> >
> > [1] C. X. Hernández, H. K. Wayment-Steele, M. M. Sultan, B. E. Husic, and V. S. Pande. Variational encoding of complex dynamics. Physical Review E, 97(6):062412, 2018.
> >
> > [2] A. A. Alemi, I. Fischer, J. V. Dillon, and Kevin Murphy. Deep variational information bottleneck. In Proceedings of the 5th International Conference on Learning Representations (ICLR), 2017.
> >
> > [3] N. Tishby, F. Pereira, and W. Biale. The information bottleneck method. In Proceedings of the 37th Annual Allerton Conference on Communication, Control, and Computing (Allerton), pages 368–377. IEEE, 1999.
> >
> > [4] Y., Jianwei, J. Lu, S. Lee, D. Batra, and D. Parikh. Graph r-cnn for scene graph generation. In *Proceedings of the European conference on computer vision (ECCV)*, 2018.
> >
> > [5] B. Tao, H. -N. Dai, J. Wu, I. W. -H. Ho, Z. Zheng and C. F. Cheang, "Complex Network Analysis of the Bitcoin Transaction Network," in *IEEE Transactions on Circuits and Systems II: Express Briefs*, vol. 69, no. 3, pp. 1009-1013, March 2022, doi: 10.1109/TCSII.2021.3127952.
> >
> > [6] J. Biagioni and J. Eriksson. Inferring road maps from global positioning system traces: Survey and comparative evaluation. Transportation Research Record, 2291(1):61–71, 2012.
> >
> > [7] C. Song, Y. Lin, S. Guo and H. Wan.Spatial- Temporal Synchronous Graph Convolutional Networks: A New Framework for Spatial-Temporal Network Data Forecasting. In Proceedings ofthe AAAI Conference on Artificial Intelligence (AAAI), volume 34, 914–921.
> >
> > [8] C. Chen, K. Petty, A. Skabardonis, P. Varaiya and Z. Jia. Freeway performance measurement system: mining loop detector data. Transportation Research Record 1748(1):96–102.
> >
> > [9] M. Maucher, B. Kracher, M. Kühl, and H. A. Kestler. Inferring Boolean network structure via correlation. Bioinformatics, 27(11):1529–1536, 04 2011.
> >
> > [10] K. Liu, H. Liu, D. Sun, and L. Zhang. Network inference from gene expression data with distance correlation and network topology centrality. Algorithms, 14(2), 2021.
> >
> > [11] Y. Zuo, G. Yu, M. G. Tadesse, and H. W. Ressom. Biological network inference using low order partial correlation. Methods, 69(3):266–273, 2014.
> >
> > [12] S. Kim. ppcor: an R package for a fast calculation to semi-partial correlation coefficients. Communications for Statistical Applications and Methods, 22(6):665, 2015.
> >
> > [13] F. Noé and F. Nuske. A variational approach to modeling slow processes in stochastic dynamical systems. Multiscale Modeling & Simulation, 11(2):635–655, 2013.
> >
> > [14] Authors Anonymous. Benchmarking structural inference methods for dynamical interacting systems. https://structinfer.github.io/, 2023.
> >
> > [15] A. Pratapa, A. P. Jalihal, J. N. Law, A. Bharadwaj, and T. Murali. Benchmarking algorithms for gene regulatory network inference from single-cell transcriptomic data. Nature Methods, 17(2): 147–154, 2020.

---

> > > ### Comment · Reviewer_5DvY · 2023-12-01
> > >
> > > Thank you for your detailed answers to our questions. The experimental results on real-world data sets prove the effectiveness of your proposed method. Your good answers to other questions also show that your work is indeed good. Therefore, after thinking about it, we decided to adjust the score to 6: marginally above the acceptance threshold

---

> ### Author Response · Authors · 2023-11-21
> **Inquiry Before Discussion Deadline**
>
> Dear Reviewer 5DvY,
>
> We hope you are doing well. As the deadline for the author-reviewer discussion phase is nearing, we would like to courteously inquire if our rebuttal has effectively addressed the concerns you raised in your review of our paper.
>
> Your insightful feedback, especially regarding experiments with real-world data, the comparison of correlation-based methods, and the assessment of time window lengths, has been invaluable in refining our work. We have endeavored to address each of these points meticulously in our rebuttal and are eager to know if our responses and subsequent revisions have met your expectations and clarified the issues you pointed out.
>
> We recognize the demands on your time and deeply appreciate any further feedback you can offer on our rebuttal. Your expertise is not only vital for the review process but also enriches our ongoing learning and growth in this field.
>
> Thank you once again for your time and invaluable insights. We eagerly await your response.
>
> Best regards,
> Authors

---

### Official Review · Reviewer_Fg5F · 2023-11-01

**Soundness:** 2 fair
**Presentation:** 4 excellent
**Contribution:** 3 good
**Rating:** 6
**Confidence:** 3

**Summary:**

This paper proposes a new method for structure inference from networked dynamical systems. At a high level, the authors' methodology is as follows. First , they use a variant of VAE's -- called a Variational Dynamics Encoder (VDE) -- to find a low-dimensional embedding of the time series of each node in the network. The network structure between nodes can then be determined by computing the partial correlation scores between the nodes' latent representations. The authors then test their methods on several benchmark datasets. Their methods perform favorably against prior approaches.

**Strengths:**

The paper is very well written, and the approach is clearly laid out. The goal of the paper -- that of learning the network that facilitates dynamic processes -- is of wide applicability and significant interest. A major strength of this work is that the proposed framework performs better than several other prior approaches for this task.

**Weaknesses:**

- There are some missing details about key elements of the procedure. How are the dynamics modeled, and how does the adjacency matrix A influence the dynamics? How is A estimated from the embedded representations? It's hinted that this is done by the partial correlations formula, but it doesn't seem to be explicitly stated.
- It's not intuitive to me why the node-level dynamics can be predicted from a single node's trajectory alone, without needing to know the dynamics of other nodes (which certainly influence the trajectories of the node in question). In other words, won't the accuracy of trajectory predictions be strongly affected if the VDE is applied separately to each node trajectory, rather than jointly?
- The authors state that scalability is a major contribution of the method. But in the Experiments, the largest network has only 250 nodes, which makes me question whether there is sufficient evidence for scalability.
- Various missing details in the Experiments section -- see the "Questions" section below.

**Questions:**

- (pg. 4) You write that $\tau$ represents the length of the time window required for the system's dynamics to be considered Markovian. What do you precisely mean by this? Why would the time window affect whether or not the system is Markovian?
- What is the dimension of the embedding of node-level dynamics produced by the VDE? And what is the effect of the dimension on the performance of the algorithm?
- It's not clear to me why VDE is used to produce an embedding, rather than another embedding procedure. Was there any particular reason for using this method?
- What are the "Springs" and "NetSim" simulations?
- What does "level of added Gaussian noise" mean? Are you changing the mean of the noise? Please be clear.

---

> ### Author Response · Authors · 2023-11-15
> **Response To Reviewer Fg5F (Part 1)**
>
> We thank Reviewer Fg5F for the constructive feedback and are pleased that our SIDEC framework's novelty and significance have been recognized. We address each of your concerns below to provide further clarity and detail.
>
> > **W1.** There are some missing details about key elements of the procedure. How are the dynamics modeled, and how does the adjacency matrix $A$ influence the dynamics? How is $A$ estimated from the embedded representations? It's hinted that this is done by the partial correlations formula, but it doesn't seem to be explicitly stated.
>
> We would like to thank the reviewer for the constructive comments and pointing out the missing details. Here are our answers:
>
> - **How are the dynamics modeled?**
>
>   As mentioned in the second paragraph of Section 4.1 in our original submission, we use the Variational Dynamics Encoder (VDE) [1], guided by the Variational Information Bottleneck (VIB) principle [2], to model dynamics. The VDE operates on the principle:
>
>   $ \mathbf{z}^t_i = \arg\min_{\mathbf{z}^t_i} I(\mathbf{z}^t_i;v^t_i)-\mathfrak{u}\cdot I(\mathbf{z}^t_i; v^{t+\tau}_i),$ where $\mathbf{z}^t_i$ is the latent variable for node $i$ at time $t$, and $\mathfrak{u}$ is the Lagrangian multiplier that balances the trade-off between sufficiency and minimality. $v^{t}_i$ and $v^{t+\tau}_i$ represent the feature of node $i$ at time $t$ and $t+\tau$, respectively. In this work, we denote $\tau$ as the length of time window. The latent variable $z^r_i$ is utilized to model the dynamics of node $i$ at time $t$ with a refer of time window $\tau$. This approach minimizes the mutual information between the latent variable $\mathbf{z}^t_i$ (for node $i$ at time $t$) and the feature vector $v^t_i$, considering a time window $\tau$. In short, this method optimizes the trade-off between capturing relevant information and minimizing redundancy.
>
> - **How does the adjacency matrix $A$ influence the dynamics?**
>
>   According to previous works on structural inference [3, 4, 5], based on the perspective of dynamical systems [6], the adjacency matrix has a crucial influence on the node dynamics, that is, the node features evolve in a observable time period $0 \leq t \leq T$, is based on the following formulation:
>
>   $ V^{t+1} \leftarrow \\{V^0, V^1, ..., V^t; \mathbf{A}\\},$ where $V^t$ denotes the all node features at time $t$, and $\mathbf{A}$ represents the adjacency matrix of the graph, which represents the structure of interaction between nodes in the dynamical system. With the widely-adopted Markovian assumption on dynamical systems, we can further simplify the previous formulation into: $V^{t+1} \leftarrow \\{V^t;\mathbf{A}\\},$ where the future features only depend on the present features and the structure of the system. We may further instantiate the formulation above, and with a scope on every node: $v_i^{t+1} = v_i^t+ \Delta \cdot \sum_ {j \in \mathcal{U}_i}f\Big(||v_i, v_j ||\_{\alpha}\Big) ,$
>   where $\Delta$ denotes a time interval, $\mathcal{U}\_i$ is the set of nodes connected with node $i$, and $f(\cdot)$ is the state-transition function deriving to dynamics caused by the edge from node $j$ to $i$, and $||\cdot , \cdot||\_\alpha$ denotes the $\alpha$-distance. In this formulation, $\mathcal{U}\_i$ is actually derived from the adjacency matrix $\mathbf{A}$. So in all three formulations, we can see that adjacency matrix $\mathbf{A}$ plays an important role in the evolving of the system, it determines which nodes have interaction and thus make the whole system to be dynamic.
>
> - **How is $A$ estimated from the embedded representations?**
>
>   We acknowledge the oversight in our initial manuscript. The matrix $\mathbf{P}$, mentioned in Section 4.3, serves as the reconstructed adjacency matrix $\mathbf{A}$. We have revised Section 4.3 in the manuscript to explicitly state this.
> -------------------

---

> > ### Author Response · Authors · 2023-11-15
> > **Response To Reviewer Fg5F (Part 2)**
> >
> > > **W2.** It's not intuitive to me why the node-level dynamics can be predicted from a single node's trajectory alone, without needing to know the dynamics of other nodes (which certainly influence the trajectories of the node in question). In other words, won't the accuracy of trajectory predictions be strongly affected if the VDE is applied separately to each node trajectory, rather than jointly?
> >
> > We appreciate your insight into the node-level dynamics. As we mentioned in the answer of previous question, the node features can be formulated as: $v_i^{t+1} = v_i^t + \Delta \cdot \sum_{j \in \mathcal{U}_i}f\Big(||v_i, v_j||\_{\alpha}\Big)$. This equation suggests that the future state of node $i$ is influenced by its current state and its interactions with connected nodes. One might assume that lacking a holistic view of the system could simplify this relationship to $v_i^{t+1}\sim v_i^t$. However, this is not the case.
> >
> > From an information-theoretic standpoint, a node's current features encapsulate information about all influencing elements over time. This can be represented by modifying the above equation to: $v_i^{t+1} \rightarrow v_i^t + \Delta \cdot \sum_{j \in \mathcal{U}_i}f\Big(||v_i, v_j||\_{\alpha}\Big)$. Here, the future state at $t+1$ inherently contains information about its own past state and its interactions with adjacent nodes. This allows us to infer connections based on a single node’s trajectory, sufficient for structural inference.
> >
> > We tested an alternative approach where all node features were fed simultaneously to the VDE. This method, however, led to scalability issues due to the fixed input size being dependent on the total number of nodes. Additionally, this approach did not yield an increase in accuracy. Moreover, in such a setup, distinguishing which latent variable corresponds to which node becomes challenging, as the calculations for all nodes are interwoven in the encoder and decoder stages. Consequently, we chose to focus on single-node trajectories, which proved to be more efficient and effective for our purposes.
> >
> > -----------------------------
> > > **W3.** The authors state that scalability is a major contribution of the method. But in the Experiments, the largest network has only 250 nodes, which makes me question whether there is sufficient evidence for scalability.
> >
> > We aim to further clarify the computational complexity associated with the structural inference methods used in our study. Unlike a linear relationship, the complexity is quadratic in relation to the number of nodes. Specifically, for directed graphs, all algorithms evaluate each possible node pair, leading to a computational complexity of $O(n^2)$, where $n$ is the total number of nodes in the graph.
> >
> > In our original submission (refer to Table 1), SIDEC has demonstrated its ability to efficiently process graphs with 150, 200, and 250 nodes, which translates to 22,500, 40,000, and 62,500 directed node pairs respectively. This is in contrast to baseline methods that struggle with graphs larger than 100 nodes. To bolster our claim of SIDEC's scalability, we conducted additional experiments using the NRI simulator [7] – a tool also employed by [3, 4, 5, 8, 9] for dataset generation. These experiments involved generating spring-ball systems with 500 and 1,000 balls, producing a total of 12,000 trajectories per graph, subsequently divided into training, validation, and test sets in an 8:2:2 ratio.
> >
> > In these experiments, only SIDEC was able to operate effectively without encountering out-of-memory (OOM) issues, unlike other VAE-based baseline methods. The resulting average Area Under the Receiver Operating Characteristic (AUROC) scores over ten runs were:
> >
> > |       | w. 500 balls      | w. 1,000 balls    |
> > | ----- | ----------------- | ----------------- |
> > | SIDEC | $ 92.8 \pm 0.04 $ | $ 90.1 \pm 0.03 $ |
> >
> > These results not only demonstrate SIDEC's capability to reconstruct the structure of large graphs but also maintain high accuracy in structural inference across varying graph sizes. We have included these additional findings in Appendix D.3 of our revised paper.
> >
> > -------

---

> > ### Comment · Reviewer_Fg5F · 2023-11-22
> >
> > Thanks to the authors for their detailed responses. A few additional comments are below:
> >
> > Response to W1:
> > From the formulation ${V^{t + 1} } \leftarrow {V^t; A} $, it's not explicitly clear to me that the network structure actually influences the dynamics, or if it does, what form the influence takes. So I would suggest also writing the assumed dynamics more explicitly.
> >
> > Response to W2:
> > While your explanation seems plausible from a conceptual standpoint, I'd suggest adding some experiments showing that it doesn't help too much to add the impact of other nodes' trajectories, and that single trajectories suffices. This would provide strong justification for your choice of treating node trajectories as "independent" in the VDE stage.
> >
> > Response to Q1:
> > Thanks for the explanation, though I feel that the way "Markovianity" is explained is confusing. If a continuous-time system is Markovian, then, from a purely mathematical standpoint, the choice of time window does not affect Markovianity. If I understand you correctly, what you really mean is that the system under consideration may not be exactly Markov, but rather approximately Markov. In such a case, time windows that are too short may still induce dependencies on past states, while larger time windows will have limited dependencies on past states.

---

> ### Author Response · Authors · 2023-11-15
> **Response To Reviewer Fg5F (Part 3)**
>
> > **Q1.** (pg. 4) You write that $\tau$ represents the length of the time window required for the system's dynamics to be considered Markovian. What do you precisely mean by this? Why would the time window affect whether or not the system is Markovian?
>
> In the context of the paper, when it is mentioned that $\tau$ represents the length of the time window required for the system's dynamics to be considered Markovian, it is referring to the concept of Markovianity in continuous-time Markov processes [10]. In these processes, the future state of the system is determined solely by its current state and not by how it arrived in that state. This property is essential to simplify the modeling and analysis of complex dynamical systems.
>
> The time window $\tau$ is crucial because it sets the scale at which the dynamics are examined. For a system to be considered Markovian, the choice of $\tau$ should be such that the memory of past states does not significantly influence the future states beyond this time window. In practical terms, this means that if you know the state of the system at time $\tau$, then this information is sufficient to predict its state at time $t+\tau$ without needing to know its earlier history.
>
> The impact of the time window $\tau$ on the Markovian nature of the system can be understood as follows:
>
> - If $\tau$ is too short, the system's evolution may still be influenced by its immediate past, not adhering to the Markovian property.
> - If $\tau$ is appropriately chosen, it allows for the dynamics of the system to 'forget' its past states, thus adhering more closely to the Markovian assumption. This is critical for the effective application of Markov models and for simplifying the analysis of the system's dynamics. This could be the case of analyzing open quantum system which somehow expresses its non-Markovian characteristic [11].
>
> Therefore, the correct selection of $\tau$ is vital to ensure that the system behaves in a Markovian manner, which is a fundamental assumption for the modeling techniques discussed in the paper. We added Appendix A.4 to state it.
>
> -----------
>
> > **Q2.** What is the dimension of the embedding of node-level dynamics produced by the VDE? And what is the effect of the dimension on the performance of the algorithm?
>
> In our original submission (Section 4.1), we detailed that the embedding dimension for node-level dynamics produced by the VDE is one. Specifically, for a given node feature $v_i^t$ (representing node $i$ at time $t$), the corresponding embedding $\mathbf{z}^t_i$ generated by the VDE also has a dimension of one.
>
> This dimensionality is a critical aspect of our methodology, as the use of the partial correlation coefficient (PCOR) requires the input dimension for each node at every time step to be exactly one. This requirement presents a limitation in directly applying PCOR for structural inference with nodes that have multiple features at any time step. In such cases, our method is constrained by the necessity to process each feature individually due to the single-dimensional nature of the embeddings.
>
> -------------------------------

---

> > ### Author Response · Authors · 2023-11-15
> > **Response To Reviewer Fg5F (Part 4)**
> >
> > > **Q3.** It's not clear to me why VDE is used to produce an embedding, rather than another embedding procedure. Was there any particular reason for using this method?
> >
> >
> > In addition to the details provided in Section 4.1 of our paper, we chose the Variational Dynamics Encoder (VDE) for the following reasons:
> >
> > 1. **Time-Series Adaptation**:  VDE is uniquely designed for time-series data, making it exceptionally adept at capturing dynamic temporal behaviors. This feature is crucial in domains such as biophysics, finance, and other fields dealing with complex dynamical systems.
> > 2. **Handling Complex Dynamics**: VDE excels at processing complex, non-linear dynamics. This capability offers a significant edge over linear dimensionality reduction techniques like PCA and tICA [1], which may not fully capture the intricacies of data in systems characterized by non-linear interactions.
> > 3. **Predictive Capability**: The time-lagged methodology of VDE enables it to forecast future states of a system based on current observations. This predictive power is invaluable for understanding and anticipating the evolution of dynamic systems, while also embedding vital dynamics information within the generated latent variables.
> > 4. **Deep Learning Integration**: As a variant of the Variational Autoencoder (VAE), VDE leverages the latest advancements in deep learning. This includes handling high-dimensional data and deriving rich, nuanced representations that might elude more traditional methods.
> > 5. **Autocorrelation Loss**: A unique aspect of VDE is its loss function, which combines reconstruction loss with autocorrelation loss. This feature is particularly effective in capturing long-term dynamics within time-series data, addressing the common issue of trajectory under-sampling in real-world applications.
> >
> > In essence, VDE's capabilities are particularly pronounced in scenarios involving complex, nonlinear, and dynamic time-series data, which are typical in structural inference of dynamical systems.
> >
> > -----------
> > > **Q4.** What are the "Springs" and "NetSim" simulations?
> >
> > The simulations utilized for generating the trajectories in the StructInfer datasets [13] are crucial for our study. Given the current unavailability of the GitHub repository and the limited documentation on the StructInfer authors' website (maybe the authors modified them to follow double-blinded review rules), we provide a brief overview of the two key simulations based on the information available:
> >
> > **Springs Simulation.** This simulation, as outlined by [7], models the motion of spring-connected particles within a 2D space. In this setup:
> >
> > - Nodes represent the particles.
> >
> > - Edges correspond to springs, governed by Hooke's law.
> >
> > - The dynamics follow a second-order ordinary differential equation:
> >
> >   $m_i\cdot x''i(t) = \sum_{j\in\mathcal{N}_i}-k\cdot\big(x_i(t)-x_j(t)\big)$, where:
> >
> >   - $m_i$denotes the particle mass, assumed to be 1.
> >   - $k$ is the spring constant, set at 1.
> >   - $\mathcal{N}_i$ represents the set of neighboring nodes connected to node $i$.
> >
> > - The equation is integrated to calculate $x'_i(t)$ and $x_i(t)$ at each time step $t$, resulting in 4D node features.
> >
> > **NetSims Simulation.** Based on the NetSim dataset [12], this simulation represents blood-oxygen-level-dependent (BOLD) imaging data from various human brain regions. Key aspects include:
> >
> > - Nodes symbolize spatial regions of interest in the brain.
> >
> > - The interaction graphs define the connections between these regions.
> >
> > - Dynamics are described by a first-order ODE model:
> >
> >   $x_i'(t) = \sigma \cdot \sum_{j\in \mathcal{N}_i}x_j(t)-\sigma \cdot x_i(t) + C\cdot u_i$, where:
> >
> >   - $\sigma$ manages temporal smoothing and neural lag, set to 0.1 based on [12].
> >   - $C$ is used to regulate external input interactions and is set to zero to minimize noise.
> >
> > - The model generates 1D node features at each time step from the computed $x_i(t)$.
> >
> > We've included detailed descriptions of the Springs and NetSim simulations in Appendix B of our revised paper.
> >
> > -----

---

> ### Author Response · Authors · 2023-11-15
> **Response To Reviewer Fg5F (Part 5)**
>
> > **Q5.** What does "level of added Gaussian noise" mean? Are you changing the mean of the noise? Please be clear.
>
> The StructInfer dataset [13], which we utilized in our study, includes trajectories with Gaussian noise added at various levels. As detailed in the dataset's documentation, the authors introduced this noise to the generated trajectories to simulate real-world conditions more accurately. The modified node features, represented as $\tilde{v}_i^t$, are calculated using the formula: $\tilde{v}_i^t = v_i^t + \zeta \cdot 0.02 \cdot \Delta$, where $\zeta$ is a standard Gaussian random variable ($\zeta \sim \mathcal{N}(0,1)$), $v_i^t$ denotes the original feature vector of node $i$ at time $t$, and $\Delta$ represents the designated noise level. The noise levels applied range from 1 to 5 across all the original trajectories.
>
> This addition of Gaussian noise is an important aspect of our experimental data, and we have updated our paper to include a comprehensive description of this process in Appendix B.
>
> We extend our gratitude to Reviewer Fg5F for the thorough, constructive, and insightful feedback. It is gratifying to know that our work has piqued your interest. We sincerely hope that our responses adequately address your concerns and contribute to enhancing the quality of our paper. To facilitate easy identification, sections and paragraphs that have been revised in the manuscript are highlighted in blue.
>
> --------------------
> ### References
>
> [1] C. X. Hernández, H. K. Wayment-Steele, M. M. Sultan, B. E. Husic, and V. S. Pande. Variational encoding of complex dynamics. Physical Review E, 97(6):062412, 2018.
>
> [2] A. A. Alemi, I. Fischer, J. V. Dillon, and Kevin Murphy. Deep variational information bottleneck. In Proceedings of the 5th International Conference on Learning Representations (ICLR), 2017.
>
> [3] A. Wang and J. Pang. Iterative structural inference of directed graphs. In Advances in Neural Information Processing Systems 35 (NeurIPS), 2022.
>
> [4] A. Wang and J. Pang. Active learning based structural inference. In Proceedings of the 40th International Conference on Machine Learning (ICML), pages 36224-36245. PMLR, 2023.
>
> [5] A. Wang, T. P. Tong and J. Pang. Effective and efficient structural inference with reservoir computing. In Proceedings of the 40th International Conference on Machine Learning (ICML), pages 36391-36410. PMLR, 2023.
>
> [6] M. Irwin and Z. Wang. Dynamic Systems Modeling, pp. 1–12. John Wiley & Sons, Ltd, 2017.
>
> [7] T. Kipf, E. Fetaya, K.-C. Wang, M. Welling, and R. Zemel. Neural relational inference for interacting systems. In Proceedings of the 35th International Conference on Machine Learning (ICML), pages 2688–2697. PMLR, 2018.
>
> [8] S. Löwe, D. Madras, R. Z. Shilling, and M. Welling. Amortized causal discovery: Learning to infer causal graphs from time-series data. In Proceedings of the 1st Conference on Causal Learning and Reasoning (CLeaR), pages 509–525. PMLR, 2022.
>
> [9] S. Chen, J. Wang, and G. Li. Neural relational inference with efficient message passing mechanisms. In Proceedings of the 35th AAAI Conference on Artificial Intelligence (AAAI), pages 7055–7063, 2021.
>
> [10] F. Noé and F. Nuske. A variational approach to modeling slow processes in stochastic dynamical systems. Multiscale Modeling & Simulation, 11(2):635–655, 2013.
>
> [11] C.F. Li, G.C. Guo, and J. Piilo. Non-Markovian quantum dynamics: What does it mean? *Europhysics Letters*, *127*(5), p.50001, 2019.
>
> [12] S. M. Smith, K. L. Miller, G. Salimi-Khorshidi, M. Webster, C. F. Beckmann, T. E. Nichols, J. D. Ramsey, and M. W. Woolrich. Network modelling methods for FMRI. Neuroimage, 54(2): 875–891, 2011.
>
> [13] Authors Anonymous. Benchmarking structural inference methods for dynamical interacting systems. https://structinfer.github.io/, 2023.

---

> ### Author Response · Authors · 2023-11-21
> **Inquiry Before Discussion Deadline**
>
> Dear Reviewer Fg5F,
>
> We hope you are doing well. As the deadline for the author-reviewer discussion phase is nearing, we would like to courteously inquire if our rebuttal has effectively addressed the concerns you raised in your review of our paper.
>
> Your insightful feedback, especially regarding the setup of the proposed method, the scalability of SIDEC, and the datasets, has been invaluable in refining our work. We have endeavored to address each of these points meticulously in our rebuttal and are eager to know if our responses and subsequent revisions have met your expectations and clarified the issues you pointed out.
>
> We recognize the demands on your time and deeply appreciate any further feedback you can offer on our rebuttal. Your expertise is not only vital for the review process but also enriches our ongoing learning and growth in this field.
>
> Thank you once again for your time and invaluable insights. We eagerly await your response.
>
> Best regards,
> Authors

---

> ### Author Response · Authors · 2023-11-22
> **Thank you for the response!**
>
> Dear Reviewer Fg5F,
>
> We extend our sincere thanks for your constructive feedback and insights, which have been instrumental in refining our paper. Please find below our detailed responses to the issues raised in your last response.
>
> **W1.**  In response to your suggestion, we would like to further elucidate the role of of the adjacency matrix $\mathbf{A}$ in our model's dynamics. The influence of $\mathbf{A}$ can be understood through the following formulation: for node $i$ in the system, its feature at time $t+1$ is given by
> $$v_i^{t+1} = v_i^t+ \Delta \cdot \sum_ {j \in \mathcal{U}_i}f\Big(||v_i, v_j ||\_{\alpha}\Big) ,$$  where $\Delta$ denotes a time interval, $\mathcal{U}\_i$ is the set of nodes connected with node $i$, and $f(\cdot)$ is the state-transition function deriving to dynamics caused by the edge from node $j$ to $i$, and $||\cdot , \cdot||\_\alpha$ denotes the $\alpha$-distance.
> Importantly, $\mathcal{U}\_i$ is derived from the adjacency matrix $\mathbf{A}$, highlighting its pivotal role in determining the interactions between nodes and, consequently, the system's dynamics.
>
> To illustrate this, let us consider the "Springs" dataset, used in our experiments. For node $i$ in this system, its 2-dimensional velocity vector at time $t$, denoted as $x_i'(t)$ is derived from the integration of its 2-dimensional accerleration vector $x_i''(t)$. This is calculated using the formula:  $$m_i\cdot x''i(t) = \sum_{j\in\mathcal{N}_i}-k\cdot\big(x_i(t)-x_j(t)\big).$$
> Here, $m_i$ represents the mass of the node, $k$ is the spring constant, and $\mathcal{N}_i$ is the set of neighboring nodes with connections to node $i$, which is determined directly by the adjacency matrix $\mathbf{A}$.
> We hope this explanation successfully clarifies the integral role of the adjacency matrix $\mathbf{A}$ in defining the dynamics within our model.
>
> ------
>
> **W2.** We appreciate your recommendation to conduct a more straightforward comparison. We have gathered experimental data to compare the performance of SIDEC with full node trajectories (SIDEC(*Full)) and single-node trajectories. The table below presents our findings:
>
>
> |       | VN\_SP\_15    | VN\_SP\_30    | VN\_SP\_50    | VN\_SP\_100    | VN\_SP\_150    | VN\_SP\_200    | VN\_SP\_250   |
> | ----- | ----------------- | ----------------- | ----------------- | ----------------- | ----------------- | ----------------- | ----------------- |
> | SIDEC(*Full) | 95.8 $\pm$ 0.02 | 96.2 $\pm$ 0.03 | 96.0 $\pm$ 0.02 | 95.0 $\pm$ 0.01 | 94.6 $\pm$ 0.03 | OOM  | OOM  |
> | SIDEC | 97.6 $\pm$ 0.01 | 97.0 $\pm$ 0.02 | 96.5 $\pm$ 0.03 | 95.7 $\pm$ 0.02 | 95.3 $\pm$ 0.02 |   95.5 $\pm$ 0.02 | 95.0 $\pm$ 0.03  |
>
> The results indicate that SIDEC(*Full) is consistently outperformed by SIDEC using single-node trajectories. Additionally, using full trajectories leads to scalability issues when facing graphs with more than 150 nodes. It also complicates the identification of latent variables corresponding to specific nodes. These findings justified our choice to focus on single-node trajectories in the VDE stage for efficiency and efficacy. We have included these results in a new section, Appendix A.5, in the revised manuscript.
>
> ------------
> **Q1.** We concur with your observation regarding the continuous nature of real processes. While the system may not be precisely Markovian, selecting an appropriate time window length can effectively limit dependencies on past states after sampling individual time steps. We have incorporated this consideration into our methodology.
>
> We are deeply grateful for your comments and suggestions, which have significantly contributed to the advancement of our research. We hope our responses have adequately addressed your concerns.
>
> Warm regards,
> Authors

---

> > ### Comment · Reviewer_Fg5F · 2023-11-22
> >
> > Thanks again for the detailed and thorough responses. In light of the added details to the manuscript, I'm happy to increase my score. Regarding W1, I would suggest explicitly adding the equation
> > $$
> > v_i^{t + 1} = v_i^t + \Delta \sum_{j \in \mathcal{U}_i} f( \| v_i, v_j \|_\alpha)
> > $$
> > to the manuscript so that the assumed model is clear.

---

> ### Author Response · Authors · 2023-11-22
> **Many thanks for the prompt and encouraging response!**
>
> Dear Reviewer Fg5F,
>
> We deeply appreciate your acknowledgment of our detailed responses and are encouraged by your decision to increase the evaluation score. Thank you for your valuable feedback and support throughout this review process.
>
> Regarding your suggestion on W1, we agree that explicitly including the equation  $$ v_i^{t + 1} = v_i^t + \Delta \cdot \sum_{j \in \mathcal{U}i} f( || v_i, v_j ||_\alpha) $$ in the manuscript will provide clarity on the model we assume. We have incorporated this equation into Section 3 to ensure the model's framework is clearly understood.
>
> Once again, we extend our gratitude for your constructive feedback and for recognizing the enhancements made to our manuscript. Your insights have been instrumental in refining our work.
>
> Warm regards,
> Authors

---

### Author Response · Authors · 2023-11-23
**Rebuttal Summary**

Dear Reviewers,

We sincerely appreciate your motivating and insightful feedback. Your recognition of our method's **novelty** (Reviewers 5DvY and LxHJ), SIDEC's **outstanding performance** (Reviewers Fg5F, 5DvY and LxHJ), its **significance** (Reviewers Fg5F and LxHJ), and the **well-structured nature** of our paper (Reviewers Fg5F, 5DvY and LxHJ) is truly encouraging. Your valuable suggestions and comments have guided us in significantly enhancing our submission. We are particularly thankful to all the reviewers for your constructive discussions, which not only improve this work but also lay the groundwork for our future research.

For ease of review, we have highlighted all revisions in blue in the updated submission. Below is a summary of the key changes:

1. **Section 4.3 Revision**: Following Reviewer Fg5F's recommendation, we clarified the obtainment process of the adjacency matrix $\mathbf{A}$ in SIDEC.
2. **Expanded Experiments**: Addressing Reviewer Fg5F's suggestion, we included additional results on networks with 500 and 1,000 nodes in Appendix D.3, to support the scalability of SIDEC.
3. **Dataset Details**: In response to Reviewer Fg5F's query, we have detailed the Springs and NetSim simulations, and the added Gaussian noise in Appendix B.
4. **Appendix A.5 Addition**: We showcase SIDEC results with trajectories of all nodes to justify our focus on single-node trajectories in the VDE stage, addressing Reviewer Fg5F's query.
5. **Real-World Dataset Performance**: In Appendix D.4, we present SIDEC's performance on real-world datasets, as requested by Reviewer 5DvY.
6. **Training Time Impact Analysis**: Appendix A.7 now includes an analysis of the impact on training time after combining VDE and PCOR, fulfilling Reviewer 5DvY's request.
7. **Algorithm Rearrangement**: We reorganized the algorithms in the Appendix for a more logical flow, per Reviewer LxHJ's suggestion.

We believe these revisions comprehensively address the concerns raised and substantially strengthen the paper.

Warm regards,

The Authors

---

### Meta-Review · Area_Chair_XUt7 · 2023-12-04

**Metareview:**

This paper combines variational dynamics encoder ideas with more classical methods using partial correlation coefficients to enable flexible structural inference. This includes multivariate data and those arising from directed graphical models. With one enthusiastic reviewer and others that lean positive after a thorough rebuttal, the paper is sufficiently supported and is better explained after incorporating reviewer remarks.

**Justification For Why Not Higher Score:**

Paper proposes a solid but not groundbreaking approach and most reviewers are borderline

**Justification For Why Not Lower Score:**

Positive consensus

---

### Decision · Program_Chairs · 2024-01-16

Accept (poster)